# Modeling spatial and temporal variability in educational development index of Bangladesh using socio-economic, and demographic data, 2001–2021: A Bayesian approach

**Afroza Sultana\*, Akher Ali, Md. Sifat Ar Salan, Mohammad Alamgir Kabir, Md. Moyazzem Hossain** \*

Department of Statistics and Data Science, Jahangirnagar University, Savar, Dhaka, Bangladesh

\* afrozasultana5161@gmail.com (AS); hossainmm@juniv.edu (MMH)

## Abstract

### Background

The Educational Development Index (EDI) is a critical tool for assessing and tracking the progress of education systems from local to national, and even global scales and needs to be chosen for every layer of the subnational boundaries to secure the basic human rights of the people. In reality, there are significant variations within the consecutive time breaks and the geographical boundaries that need to be examined. The authors aim to examine how the EDI relates to various spatiotemporal variables.

### Methods and Materials

This research is based on secondary data on literacy rates (EDI) from 64 districts of Bangladesh and 6 relevant variables over the period 2001 to 2021. The optimal model for the data was identified from Bayesian spatial-temporal modeling (Linear, Analysis of Variance (ANOVA), Autoregressive (AR1), and AR2) and the Markov Chain Monte Carlo (MCMC) method used to generate data about the prior and posterior realizations. To select the best model different model selection and validation criteria such as the Deviance Information Criterion (DIC), Watanabe-Akaike information criterion (WAIC), and Root Mean Square Error (RMSE) were employed in this study.

### Results

The 'AR1' model is a 'temporal model' performed better than others. Significant spatial ($\rho_S$ =0.994) and temporal ($\rho_T$ =0.347) variations were identified for the suited model. Of the factors considered for model fitting, the health index, income index, expected years of schooling, population density, and dependency ratio are found to be important components of educational development in Bangladesh.

**Data availability statement:** All Data are available in the website of Global Data Lab (https://www.globaldatalab.org) and Bangladesh Bureau of Statistics (https://bbs.gov.bd/). Data file also attached as a supporting information.

**Funding:** The author(s) received no specific funding for this work.

## Conclusion

The variation in the spatial domain can be used to identify the districts to improve the educational index controlling responsible factors by the policymakers.

## Introduction

Education is a fundamental pillar of economic and social development, playing a vital role in shaping the sustainable and digitalized future of nations. It provides insights into the country's progress and well-being in various dimensions, including economic, social, health, and environmental factors [1]. Education fosters the development of human skills, which are viewed as human capital since they enable the community to access the services it requires [2]. Better educational opportunities combined with a trained workforce indicate better opportunities for economic growth and development in general. According to international and regional human rights law and other international texts, such as the Universal Declaration of Human Rights [3] education is acknowledged as one of the essential human rights [4]. Increased labor market involvement, better child and family health, improved nutrition, a reduction in poverty, and increased life possibilities are all benefits of education. The Educational Development Index (EDI) serves as an overall measure that includes a variety of educational indicators, reflecting the overall quality and accessibility of education within a particular region or country. In the context of Bangladesh, a developing country striving for progress with an area of 147,570 square kilometers and a human population density of 1,156.84 per square kilometer [5], understanding temporal and spatial variations in the EDI is crucial. Analyzing the spatiotemporal distribution of EDI in Bangladesh can provide insight into the performance of the educational system, identify areas for improvement, and suggest the allocation of resources to ensure there are relationships across sectors. Bangladesh's remarkable development journey has turned it from a nation affected by war into a growing and promising economy, often referred to as the "Bangladesh Miracle" [6]. Bangladesh has increased efforts to improve the quality of education because it recognizes that education is essential to achieving the desired degree of national development. The nation is committed to achieving the Sustainable Development Goals (SDGs) related to focusing on ensuring inclusive, equitable, and high-quality education while promoting lifelong learning opportunities for everyone, along with the associated targets. The nation has successfully achieved gender parity in alignment with the Millennium Development Goals. [4]. In Bangladesh, the government, non-governmental organizations (NGOs), and international agencies have been working tirelessly to put these words into action by improving the quality of education. Bangladeshi policymakers believe that with quality education, our population may be transformed into a useful resource for national development [7]. Despite constitutional obligations for balanced growth, regional disparities persist in Bangladesh. However, the government of our nation must address the idea of balanced regional growth because it is required under the constitution [8].

A previous study highlighted the use of geodemographics in assessing potential education inequality by utilizing a spatial approach that is demonstrated in a case study of central Beijing [9]. A community's educational attainment is a key component of its social and economic development and a predictor of future progress [10]. However, the quality, equity, and appropriateness of accessibility as well as the spatial distribution of the facilities produce differences in the efficiency of education which is proved by Bulti [11]. Regional variations exist in the literacy rate, expected years of schooling, health index, income index, child health, life expectancy, and so on spatial patterns and autocorrelation play a key role in spatial models that

potentially aid decision-makers [12]. The Bayesian Hierarchical Spatial-temporal model, rooted in the Bayesian framework, is employed to analyze the fluctuations within spatial regions and their temporal evolution. This model effectively integrates both spatial and temporal dimensions, making it a valuable tool for modeling social data [13], given the significant variability in development across different periods and geographical locations. The social and economic growth of a nation is significantly influenced by its level of education, and it is important to consider the regional variations in this regard when formulating policy recommendations. Therefore, the authors aim to examine how the EDI relates to various spatiotemporal variables in Bangladesh. The authors believe that the findings will offer insightful information to scholars examining the influence of different social factors on global educational growth.

## Materials and methods

### Study area and data

All districts of Bangladesh are considered as the spatial domain in this study. In addition, there were 64 different district names in the spatial dimension, and the temporal dimension covered 21 years in a row in these districts. The aggregate data for the 64 districts throughout the 21 years is represented by the 1344 observations (rows) for each variable. Table 1 shows the variables and their source from where the data were extracted. All the data were extracted from the Global Data Lab (https://www.globaldatalab.org) [14], and the Bangladesh Bureau of Statistics (BBS) [5] over the period 2001 to 2021. The shapefile of Bangladesh map are downloaded from the website of Humanitarian Data Exchange and can be downloaded it from the link given below https://data.humdata.org/dataset/cod-ab-bgd.

### Study variables

The EDI, also referred as "literacy rate", is a composite measure of educational outcomes. In this study, we have considered the percentage of literacy rate in each district for the predefined period as our response variable. Based on some previous studies we have considered income index ( $X_1$ ), health index ( $X_2$ ), and expected years of schooling ( $X_3$ ) as our covariates to justify our objectives [8,15–18]. Population density per 100 kilometers ( $X_4$ ), distance (in kilometers) from Dhaka (the capital of Bangladesh, $X_5$ ), and dependency ratio ( $X_6$ ) are also used as covariates for our study which were found in a similar study [19–21] (M. Ahmed & Rahamāna, 2013; Kabir & Akter, 2014; Mazumdar, 2005).

### Exploratory data analysis (EDA)

Exploratory Data Analysis (EDA) was carried out to summarize the main characteristics of a dataset with the help of graphical representations [22].

Table 1. Moran's *I* and Geary's *C* Indicators for Literacy Rate.

| Summary Statistics | Global Moran's *I* | Geary's *C* |
|---|---|---|
| Statistic | 0.486 | 0.542 |
| Expectation | -0.0158 | 1.000 |
| Variance | 0.0067 | 0.0083 |
| Standard Deviate | 6.141 | 5.0174 |
| *p* -value | $4.103 \times 10^{-10}$ | $2.619 \times 10^{-7}$ |

## Spatio-temporal modeling

We have fitted different types of separable and non-separable models to describe the observed variation in literacy rate in relation to socio-demographic variables over space and time. The response variable literacy rate, $Y_{it}$, ($i = 1,2,...,64$ and $t = 1,2,...,21$) was assumed to follow the Gaussian distribution $Y_{it} \sim N(\mu_{it}, \sigma_{it}^2)$ with probability density function,

$$f\left(y_{it} \mid \mu_{it}, \sigma_{it}^2\right) = \frac{1}{\sqrt{2\pi\sigma_{it}^2}} e^{-\frac{1}{2\sigma_{it}^2}(y_{it} - \mu_{it})^2}, \mu_{it} \in \mathbb{R}, y_{it} \in \mathbb{R}$$

where, $\mu_{it}$ is the mean and $\sigma_{it}^2$ is the variance of the defined response variable at space-time combination.

Let us say that $x_{it}$ represents the covariates of $p$-dimensional at the space-time combination. The model can be written as

$$Y_{it} \sim x_{it}'\beta + \psi_{it} + \epsilon_{it} \qquad i = 1,...,n, \qquad t = 1,..., T$$

where $\varepsilon_{it} \sim N(0, \nu^2)$ independently and $\psi_{it}$ are spatio-temporal random effects, and $\nu^2$ is the error variance.

For independent error general linear regression model, $\beta$ is a matrix of unknown regression coefficients corresponding to each covariate ($X$), and $\psi_{it}$ is the error term that we assume to follow $N(0, \sigma^2)$. Based on different assumptions about their construction, the random $\psi_{it}$ is considered to have the following varied range of models. To extend the process and parameter models to both space and time dimensions, several models have been used in research [23]. Knorr-Held proposed an analysis of the variance-type model with or without the inclusion of the space-time interaction [24].The general spatial-temporal models were considered as,

$$\psi_{it} = \begin{cases} \beta_1 + \varphi_i + (\beta_2 + \delta_i)\dfrac{t - \bar{t}}{T}, & \text{Linear model of trend} \\[2ex] \varphi_i + \delta_i + \gamma_{it}, & \text{Analysis of Variance}(\text{ANOVA})\text{model}. \\[2ex] \varphi_i + \delta_t, & \text{Auto Regressive model} \end{cases}$$

In the linear model of trend (Bayesian model), the overall intercept ($\beta_1$) and the slope ($\beta_2$) parameters have a flat prior distribution. The incremental intercept ($\phi_i$) and slope parameters ($\delta_i$) unique to the $i^{th}$ area. A modified Conditional Autoregressive (CAR) prior distribution is applied to both sets of parameters, and different values of $\rho$ and $\tau^2$ are used [24]. The parameters $\phi = \phi_1,...,\phi_{64}$ and $\delta = \delta_1,...,\delta_{64}$ are assigned the NCAR($\phi \mid \rho_{int}, \tau^2_{int}, W$) and NCAR($\delta \mid \rho_{slo}, \tau^2_{slo}, W$) distribution. Here $\rho_{int}$, $\rho_{slo}$, $\tau^2_{slo}$, $\tau^2_{slo}$ are autoregression and variance parameters for the intercept ($\phi_i$) and slope ($\delta_i$) processes. The variance parameters, $\tau^2_{int}$ and $\tau^2_{slo}$, follow inverse gamma prior distributions, whilst the parameters $\rho_{int}$ and $\rho_{slo}$ are assigned independent uniform prior distributions inside the unit interval (0,1). In the ANOVA model, the three sets of parameters $\phi_i$, $\delta_i$ and $\gamma_{it}$ are all random effects, follow NCAR prior distributions. Moreover, $\gamma_{it}$ is the interaction effect captures variations that cannot be explained by the spatial or temporal effects alone, assumed tthat o be independent and normally distributed for all values of $i$ and $t$ with variance $\tau^2_I$. The parameters $\rho_S$ and $\rho_T$ are assigned independent uniform prior distributions over the range (0,1), while the variance parameters $\tau^2_S$, $\tau^2_T$ and $\tau^2_I$ are modeled using inverse gamma prior distributions [25].

The models have been implemented as the "ST.CAR" linear model in the package [25]. The argument model="linear" for linear, model = "anova" for ANOVA and model = "ar" was

used for one order autoregressive (where, $\delta_t = 0$ ) and the autoregressive model of order two, AR = 2 argument was used in this study for the autoregressive model, and the "CarBayesST" package chooses this model in the "bmstdr" package. The "ggplot2" package in RStudio was used to visualize the variables. The shape files were also read using the "readOGR" function. The R programming language and the "Bcartime" package were used for geographical analysis and model fitting [25]. To select the appropriate model, Bayesian model selection criteria such as the Log Marginal Predictive Likelihood (LMPL), Deviance Information Criterion (DIC) [26] and Watanabe-Akaike Information Criterion (WAIC) [27] were used in this study along with RMSE, and MAE was used for evaluating the accuracy of the best model. After 20,000 burn-in iterations, all of the models were run for 120,000 iterations. To reduce autocorrelation in the MCMC setting, the data were kept following a thinning of 10 iterations.

## Assumption checking

An important assumption of spatio-temporal modeling is the existence of spatio-temporal autocorrelation. Here, autocorrelation in spatio-temporal data has been analyzed using Moran's *I* (Global and Local Moran's *I*) and Geary's *C* [28]. Although in theory Geary's coefficient and Moran's index are comparable, in actuality Geary's coefficient is based on a sample and Moran's index is based on the population [29,30]. In this study, spatio-temporal autocorrelation checking has been done as an extension of Moran's *I* and Geary's *C* [31]. To examine autocorrelation with Global Moran's, we set the null hypothesis as, $H_0$ : There is no spatial dependence between neighboring districts due to educational development, and the alternative hypothesis $H_1$ : There is spatial dependence between neighboring districts due to educational development.

## Ethical statements

It is not required because the dataset does not contain any human information and all data is publicly available.

## Results

A time series plot [Fig 1] of literacy rates in Bangladesh's 64 districts reveals a consistent increase in literacy over the past three decades.

In 2020, Dhaka district had the highest literacy rate due to its role as the education hub. Conversely, Habiganj showed the lowest literacy rate between 2001 and 2021, with Mymensingh, Kishoreganj, Sherpur, and Cox's Bazar also lagging in literacy levels, particularly in the southeast region [Fig 2].

Fig 3 shows the correlation matrix describing the relationships between literacy rate and socio-economic variables. The analysis of literacy rate and related variables reveals several key findings. There is a positive association between literacy rate and variables such as health index (0.77), income index (0.85), expected years of schooling (0.94), and population density (0.19). Wealth and health positively influence literacy rates. However, there is only a weak positive correlation between population density and literacy rate, mainly seen in urban areas. Conversely, a high dependency ratio (-0.77) is strongly negatively correlated with the literacy rate, indicating that as dependency increases, literacy tends to decrease. The distance from Dhaka to different districts shows no significant correlation with the literacy rate in the country [Fig 3].

Additionally the variation in important covariates by spatial domain are shown in Figs 4–9.

The plots illustrate significant regional disparities in Bangladesh. Dhaka and the southeastern, and northeastern regions exhibit better health outcomes, while the coastline regions have

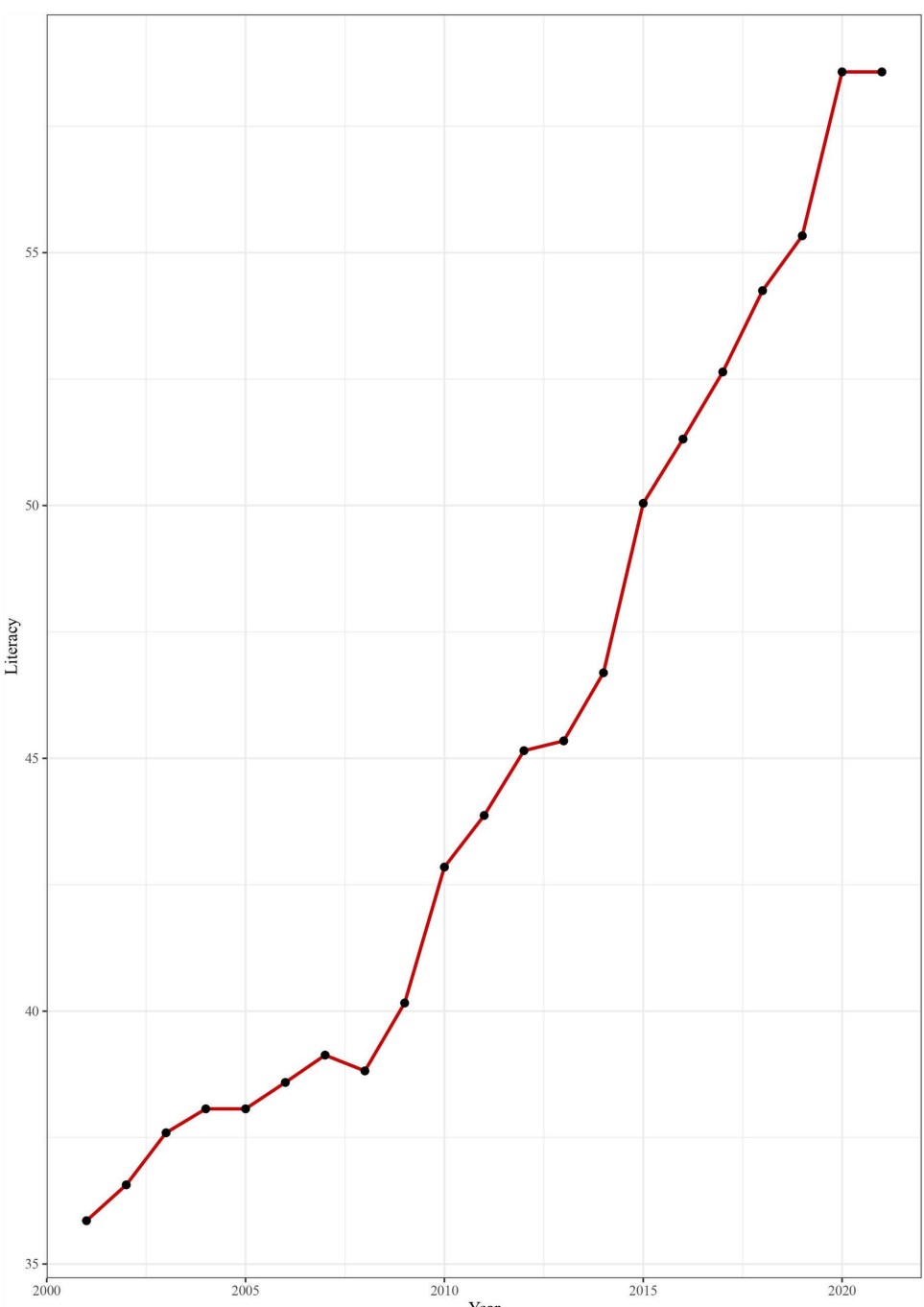

**Fig 1. Time series plot of Literacy rate in Bangladesh.**

fewer health facilities. Most districts have below-standard living conditions, with high-income indices mainly seen in Dhaka and a portion of the northwest. Barishal, Jhalokati, and Pirojpur show higher predicted years of schooling attendance. Dependency ratios are lower in Dhaka and negatively correlated with literacy. The population density is highest in Dhaka, contrasting with the mountainous region's lower density, and the northeastern districts are distant from the capital [Fig 4 - Fig 9].

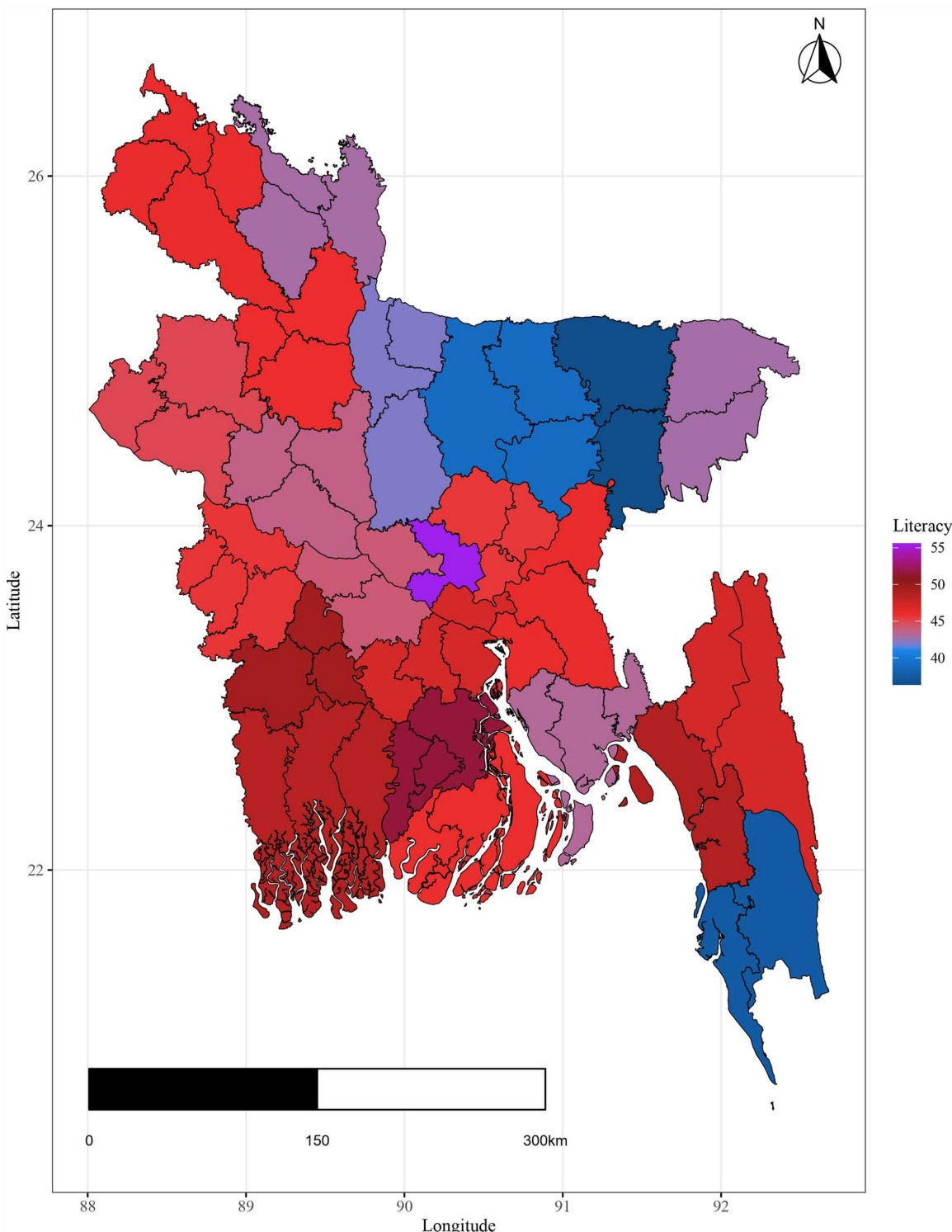

**Fig 2. Spatial distribution of Literacy rate in Bangladesh.**

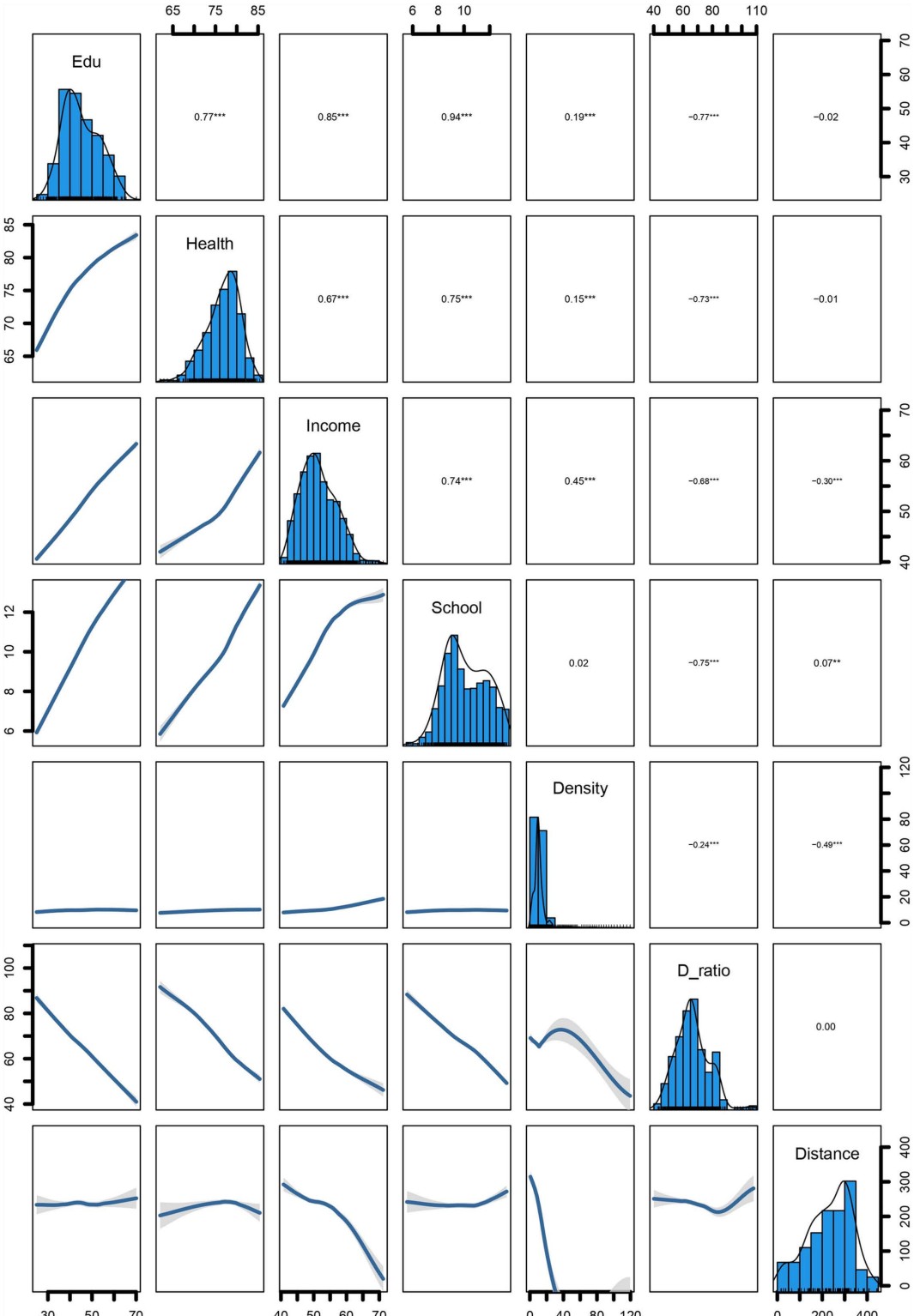

**Fig 3. The correlation matrix of the selected variables, \*\*\* presents _p_ -value < 0.001, \*\* presents _p_ -value < 0.05, \* presents _p_ -value < 0.1.**

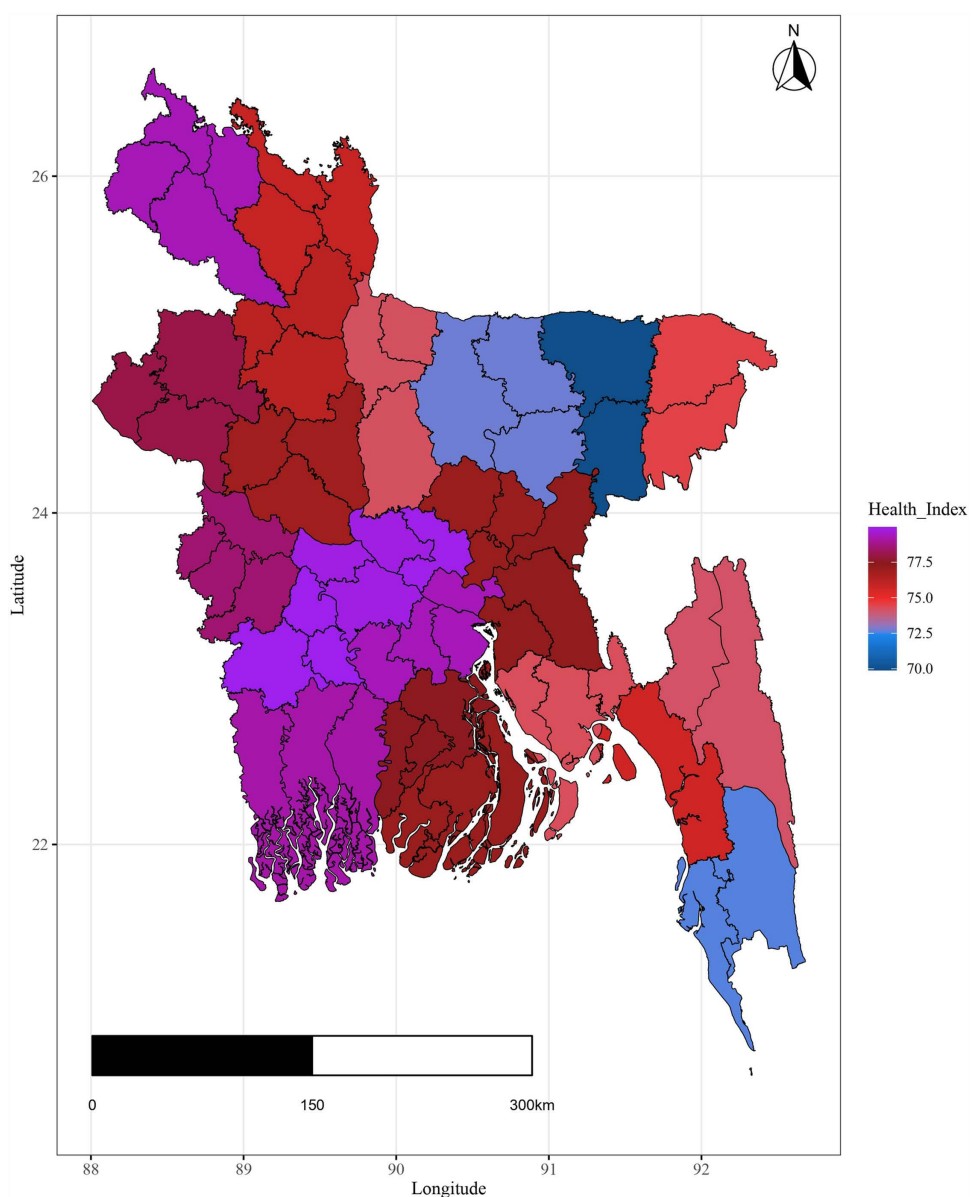

**Fig 4. District-wise variation of spatial variation of health index.**

Table 1 presents the Global Moran's *I* and Geary's *C* statistics for literacy rate as the development indicator. Moran *I*'s yielded a score of 0.486, indicating a considerable spatial autocorrelation between regional domains. The null hypothesis should be rejected, as indicated by the obtained $p$-value of $4.103\times10^{-10}$, which is less than 0.05. Moran' *I* demonstrate the $p$-value of $5\times10^{-5}$, at a 5% threshold of significance using Monte Carlo (MC) simulation of the 19999 global Moran. The null hypothesis should be disregarded in both scenarios.

A density plot, Fig 10 of the Monte Carlo permutation results show permutation outcomes on a curve and the Moran's *I* value–the latter being so far to the right of the curve, at a value of 0.486, indicates existence of positive spatial autocorrelation. A positive autocorrelation between districts is also shown by the derived Geary's *C* statistic value of 0.542, which is inside

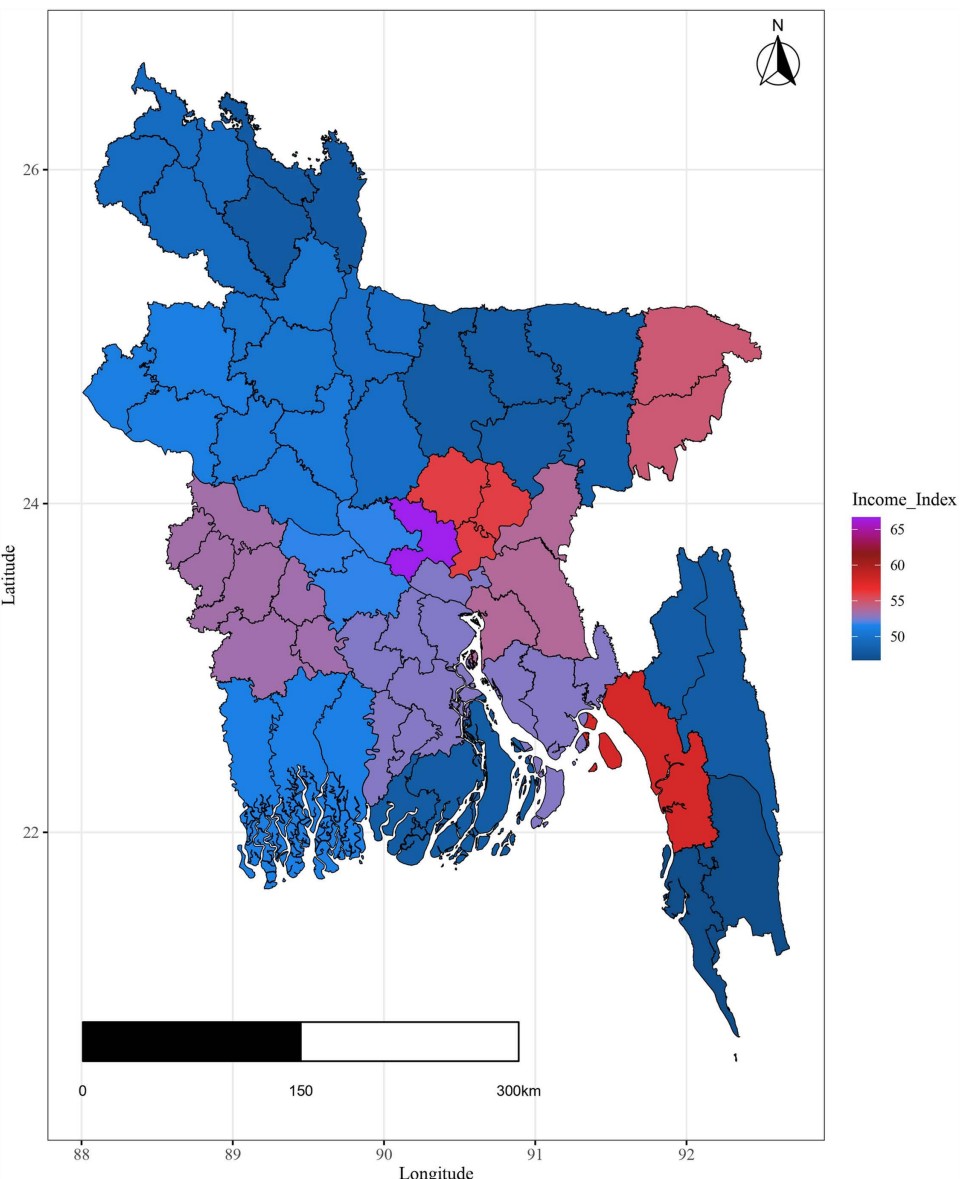

**Fig 5. Spatial variation of income index.**

the [0,1) interval. The above two spatial autocorrelation analysis methods reveal the presence of clusters between districts' literacy percentages.

To visualize the spatial autocorrelation in the dataset we constructed a "Moran scatter plot" which displays the spatial data against its spatially lagged values in Fig 11. According to the global evaluation of the dataset (positive Moran's $I$ value), there is spatial autocorrelation and it must be reduced in the MCMC setting in the modeling section.

Similar to the global analysis (above), local Moran's $I$ base the test statistic on the spatial weights of the items. While the traditional Moran's $I$ statistic offers a valuable measure of global spatial autocorrelation in illiteracy rates, it falls short of capturing local spatial patterns. Fig 12 demonstrates that such districts have low educational development along with the low neighboring districts.

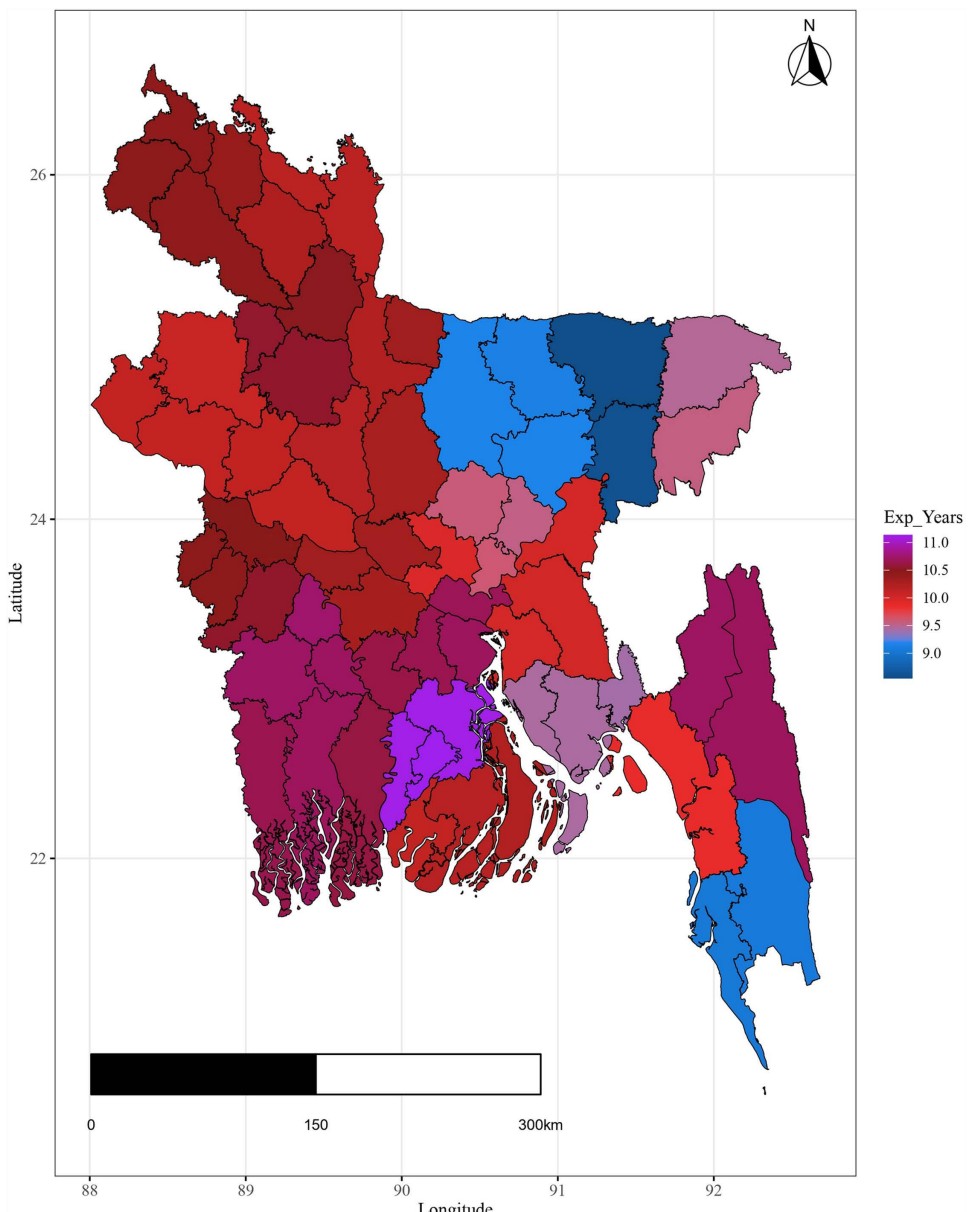

**Fig 6. Spatial variation of schooling years.**

The following districts are part of the low-high spatial cluster and meet the criteria for statistical significance: Noakhali, Manikganj, Faridpur, Feni, Lakshmipur, and Rajbari. It would seem that these districts have a low literacy rate, while it would also seem that the districts next to them have a high literacy rate. The brown color code in the graph indicates that 10 districts give the impression of having a high literacy rate. Moreover, 27 districts are in the high-high cluster category, indicating that they have neighboring districts containing a high educational development. So, a clear concept of clustering is found in the above graphical representation.

Table 2 shows the results of fitting all five models: parameter estimates and credible intervals. This table also contains different spatial and temporal indicator values for the

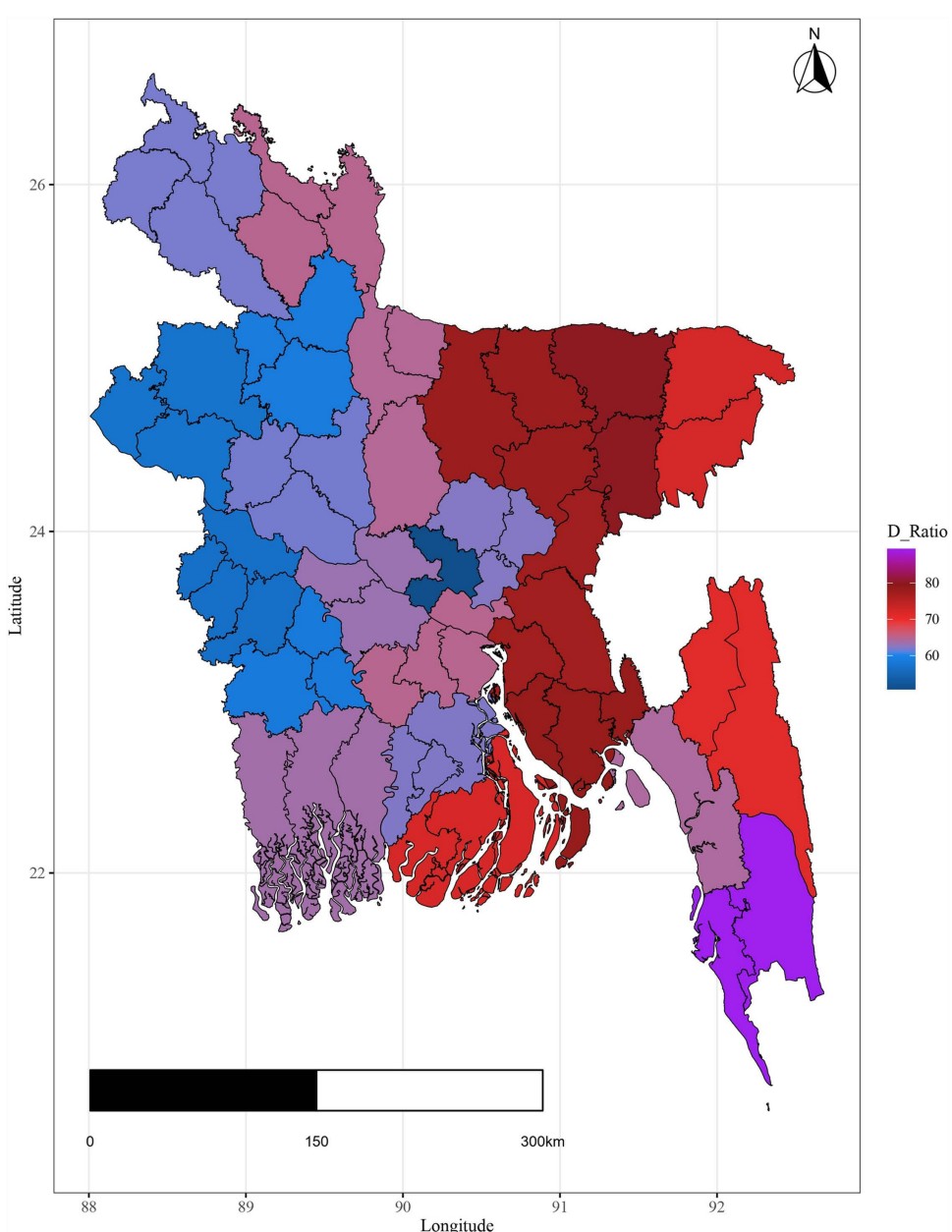

**Fig 7. Spatial variation of dependency ratio.**

four spatio-temporal models, described above. The analysis reveals the significant impact of expected years of schooling on literacy rate, across all models. Additionally, health and income indices exhibit considerable influence on literacy rates. However, the variable "distance" exerts a relatively minor effect on literacy rates, as indicated by the credible intervals in Table 2. The dependency ratio is also insignificant which matches with the pair-wise scatter plot. Thus, all values provide insights into the relationships between the predictor variables and the response variable. The estimate of the error variance $\nu^2$ is 3.919 is higher for Independent Error Bayesian GLM than the spatio-temporal models, as the spatial and temporal variance is not accounted for in the model. In the case of spatio-temporal linear models, $\tau^2_{int} = 0.586$ signifies a lower variance for intercept than for slope, $\tau^2_{slo} = 0.777$. An estimated spatial correlation

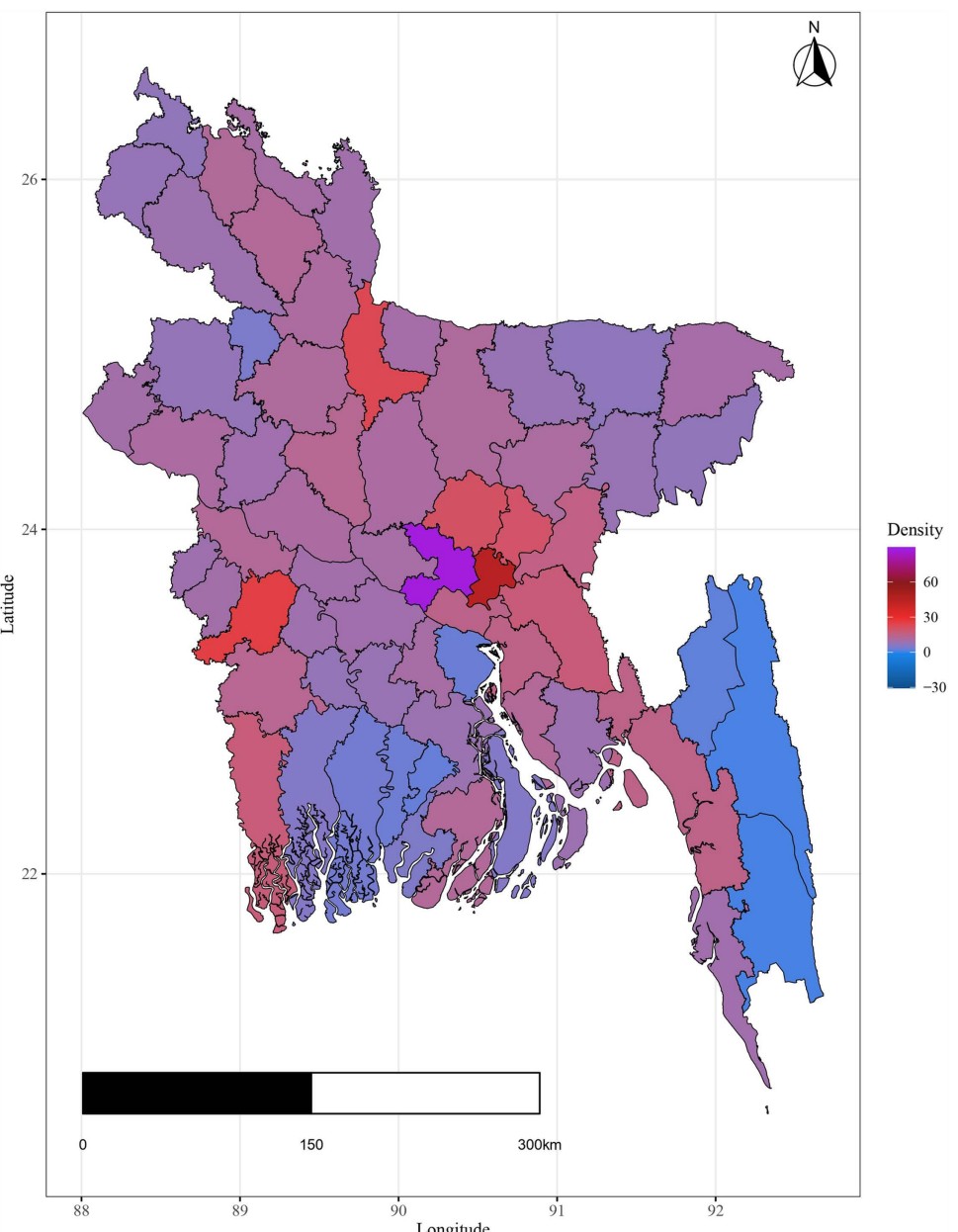

**Fig 8. Spatial variation of population density.**

parameter, $\rho = 0.693$, suggests strong evidence of spatial correlation, further supported by $\rho_{int}$ = 0.586, indicating higher auto-regression for the intercept compared to $\rho_{slo} = 0.412$. Top of FormThe ANOVA model in Table 2 displays a spatial correlation value of $\rho_S$ is 0.576, indicating significant spatial influence, and is higher than the value of $\rho_T = 0.419$ which indicates that spatial variation of literacy rate is strongly dependent on temporal effect.

The values of $\tau^2_S$ and $\tau^2_T$ for the ANOVA model in Table 2 is non-zero, which supports the necessity of taking the spatial and temporal domain into account in the model as well as the existence of geographical and temporal variation in the literacy percentage. According to estimates of the AR(1) model, the temporal auto-correlation parameter $\rho_T = 0.347$, which is

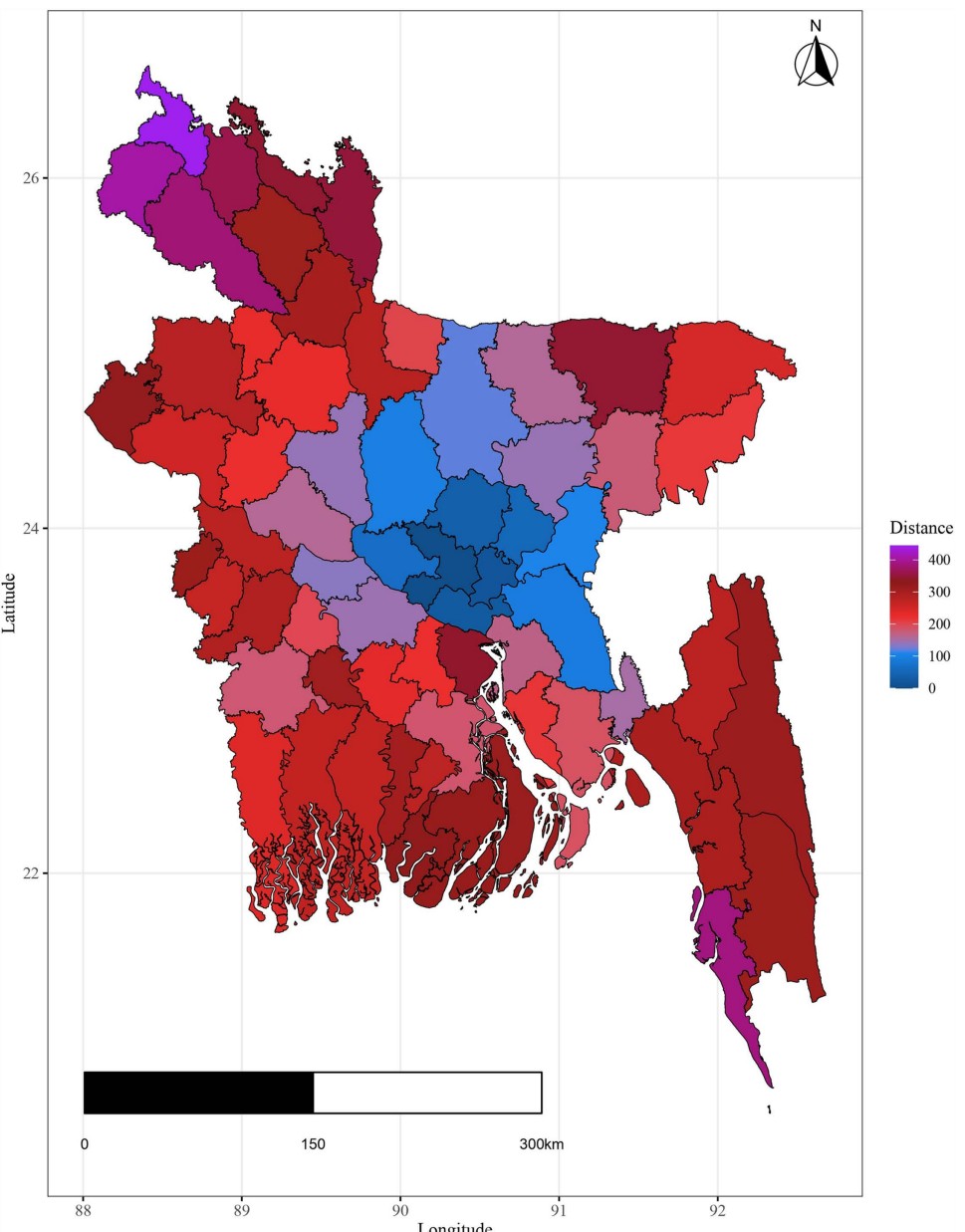

**Fig 9. Distance from Dhaka.**

roughly three times smaller than the spatial correlation value $\rho_s = 0.994$. which indicates a high spatial correlation exists in the model. Both the AR parameters of the AR(2) model show a negative lagged correlation which can be interpreted from the confidence interval of the estimated value. Thus, from above it is clear that both spatial and temporal effects of variables on literacy percentage in Bangladesh.

Table 3 combines all performance measures of the selected models to choose the best one based on model selection criteria and validation criteria. Fig 13 and Fig 14 also presents model comparison plots of criteria which is performed using the MCMC sample.

**Table 2. Estimated parameters and credible intervals for all proposed models.**

| | Independent Error Bayesian GLM | Bayesian Spatio-Temporal Model | | | |
| --- | --- | --- | --- | --- | --- |
| | | Linear | ANOVA | AR (1) | AR (2) |
| | Estimate | Estimate | Estimate | Estimate | Estimate |
| | CI | CI | CI | CI | CI |
| Intercept | −21.369 | −18.810 | −16.283 | −16.247 | −16.164 |
| | (−25.233, −17.468) | (−22.716, −14.928) | (−20.131, −12.395) | (−20.054, −12.392) | (−19.971, −12.317) |
| Health Index | 0.106 | 0.095 | 0.049 | 0.049 | 0.047 |
| | (0.061, 0.150) | (0.050, 0.139) | (0.006, 0.092) | (0.006, 0.091) | (0.005, 0.090) |
| Income Index | 0.500 | 0.487 | 0.544 | 0.543 | 0.543 |
| | (0.460, 0.541) | (0.446, 0.528) | (0.503, 0.585) | (0.502, 0.583) | (0.502, 0.584) |
| Years of Schooling | 3.200 | 3.189 | 3.053 | 3.061 | 3.062 |
| | (3.063, 3.342) | (3.051, 3.330) | (2.917, 3.188) | (2.926, 3.192) | (2.932, 3.196) |
| Population Density | 0.027 | 0.024 | 0.021 | 0.021 | 0.021 |
| | (0.015, 0.040) | (0.011, 0.036) | (0.008, 0.033) | (0.008, 0.033) | (0.008, 0.034) |
| Dependency Ratio | −0.017 | −0.030 | −0.035 | −0.035 | −0.035 |
| | (−0.034, −0.001) | (−0.047, −0.013) | (−0.052, −0.019) | (−0.051, −0.019) | (−0.051, −0.019) |
| Distance | 0.004 | 0.004 | 0.003 | 0.003 | 0.003 |
| | (0.003, 0.005) | (0.003, 0.005) | (0.002, 0.004) | (0.002, 0.004) | (0.002, 0.004) |
| $\tau^2_S$ | – | – | **0.159** | – | – |
| | | | (0.084, 0.285) | | |
| $\tau^2_T$ | – | – | **0.933** | 0.290 | **0.281** |
| | | | (0.500, 1.690) | (0.139, 0.496) | (0.133, 0.512) |
| $\nu^2$ | 3.919 | **3.721** | **3.114** | 2.988 | 2.983 |
| | (3.627, 4.233) | (3.442, 4.013) | (2.890, 3.361) | (2.747, 3.244) | (2.754, 3.248) |
| $\rho_S$ | – | – | **0.576** | **0.994** | **0.994** |
| | | | (0.170, 0.937) | (0.986, 0.998) | (0.987, 0.998) |
| $\rho_T$ | – | – | **0.419** | **0.347** | – |
| | | | (0.036, 0.855) | (0.052, 0.638) | |
| $\alpha$ | – | **-1.353** | – | – | – |
| | | (-1.724, -0.984) | | | |
| $\tau^2_{int}$ | – | **0.161** | – | – | – |
| | | (0.085, 0.290) | | | |
| $\tau^2_{slo}$ | – | **0.777** | – | – | – |
| | | (0.189, 2.181) | | | |
| $\rho_{int}$ | – | 0.586 | – | – | – |
| | | (0.169, 0.939) | | | |
| $\rho_{slo}$ | – | 0.412 | – | – | – |
| | | (0.016, 0.925) | | | |
| $\rho1_T$ | – | – | – | – | 0.247 |
| | | | | | (-0.131, 0.625) |
| $\rho2_T$ | – | – | – | – | -0.082 |
| | | | | | (-0.416, 0.277) |

**Note:** CI: Credible interval

Findings presented in Table 3, Fig 13 and Fig 14 revealed that the AR (1) model out-performs others with the lowest DIC and RMSE values. Additionally, AR (1) exhibits the highest log marginal predictive likelihood (LMPL) and log-likelihood values compared to

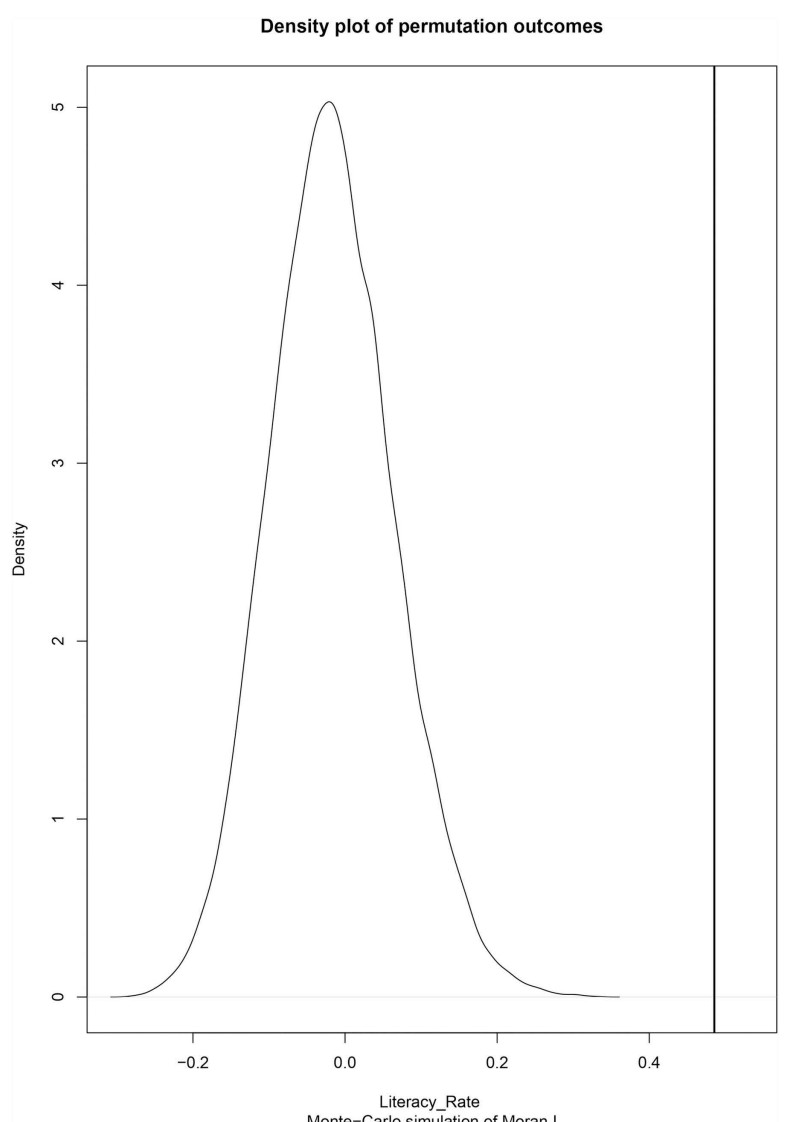

**Fig 10. Density plot of Global Moran's *I*.**

Table 3. Performance measures of the selected models.

| Models | Model Selection Criteria | | | | | | Model validation Criteria | | | |
|---|---|---|---|---|---|---|---|---|---|---|
| | DIC | p. d | WAIC | p. w | LMPL | loglikelihood | RMSE | MAE | CRPS | CVG |
| **IEBGLM** | 5659.89 | 7.963 | 5661.44 | 9.365 | -2830.72 | -2822.000 | 1.946 | 1.532 | 1.179 | 94.029 |
| **Linear** | 5619.97 | 36.69 | 5622.25 | 37.89 | -2811.23 | -2773.340 | 1.861 | 1.465 | 1.036 | 94.776 |
| **AR (1)** | **5375.25** | **88.25** | **5379.65** | **87.05** | **-2690.53** | **-2599.371** | **1.770** | **1.388** | **0.936** | **94.029** |
| **AR (2)** | 5377.82 | 85.49 | 5382.38 | 84.78 | -2691.93 | -2602.584 | 1.773 | 1.391 | 1.059 | 94.029 |
| **ANOVA** | 5388.86 | 45.73 | 5391.64 | 46.75 | -2695.91 | -2648.663 | 1.735 | 1.395 | 1.093 | 94.029 |

Note: DIC: Deviance Information Criteria; p.d: Penalty of Deviance Information Criteria; WAIC: Watanabe Akaike Information Criteria; p.w: Penalty of Watanabe Akaike Information Criteria; LMPL: Log Marginal Likelihood; RMSE: Route Mean Square Error; MAE: Mean Absolute Error; CRPS: Continuous Ranked Probability Score; CVG: Convergence

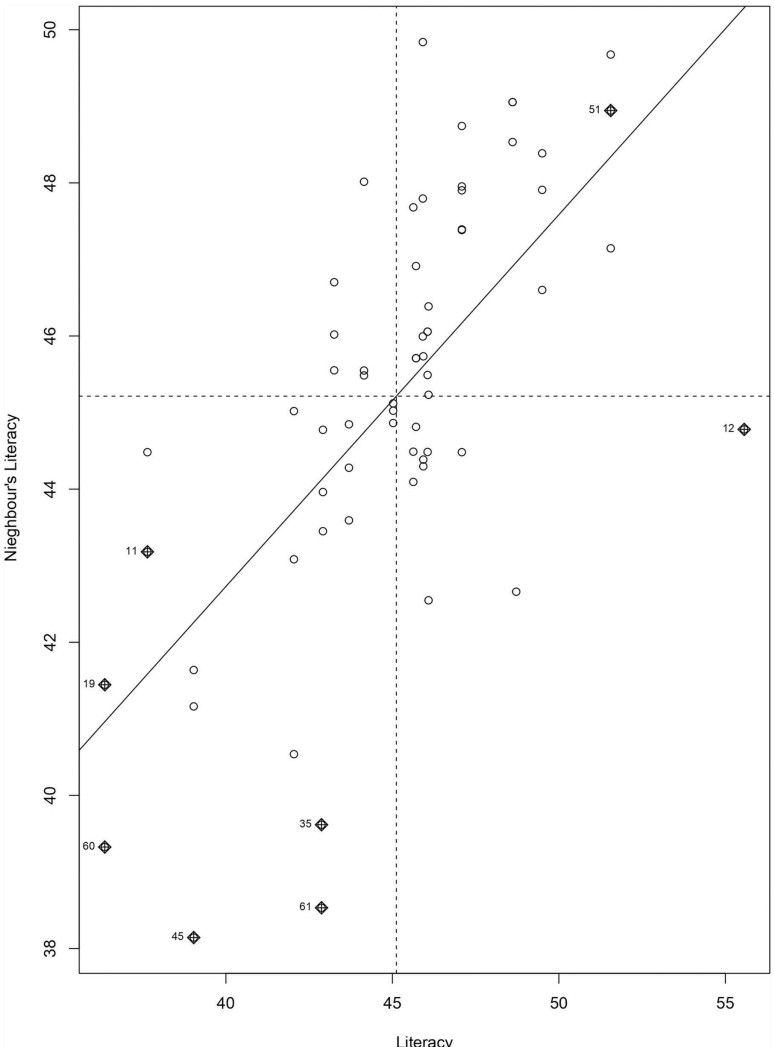

**Fig 11. Graphical representation of spatial autocorrelation.**

other spatiotemporal models. Model validation criteria further support AR (1) with lower RMSE, MAE, and CRPS. Notably, the spatio-temporal linear model demonstrates the highest convergence rate. Consequently, AR (1) emerges as the best-fitting model for data analysis.

For each observed response, the MCMC iteration generates fitted values, and from fitted values residuals are computed for fitted plot construction. The spatial residuals and standard deviation of the residuals are then calculated for each spatial domain (district) based on the range of the temporal domain. Here, 10000 samples from the MCMC are used to calculate the residuals and their standard deviation and presented in Fig 15 and Fig 16.

Fig 15 and Fig 16 display a residual plot for each district as well as a plot for the residual's standard error. It is evident from the standard error plot that the residual values do not vary excessively. The plots in Fig 15 and Fig 16 do not reveal any noteworthy spatial patterns that call for further research. Instead, the residuals from the spatio-temporal Autoregressive (AR(1)) model are represented by the spatially aggregated residual and the standard deviation of the residuals.

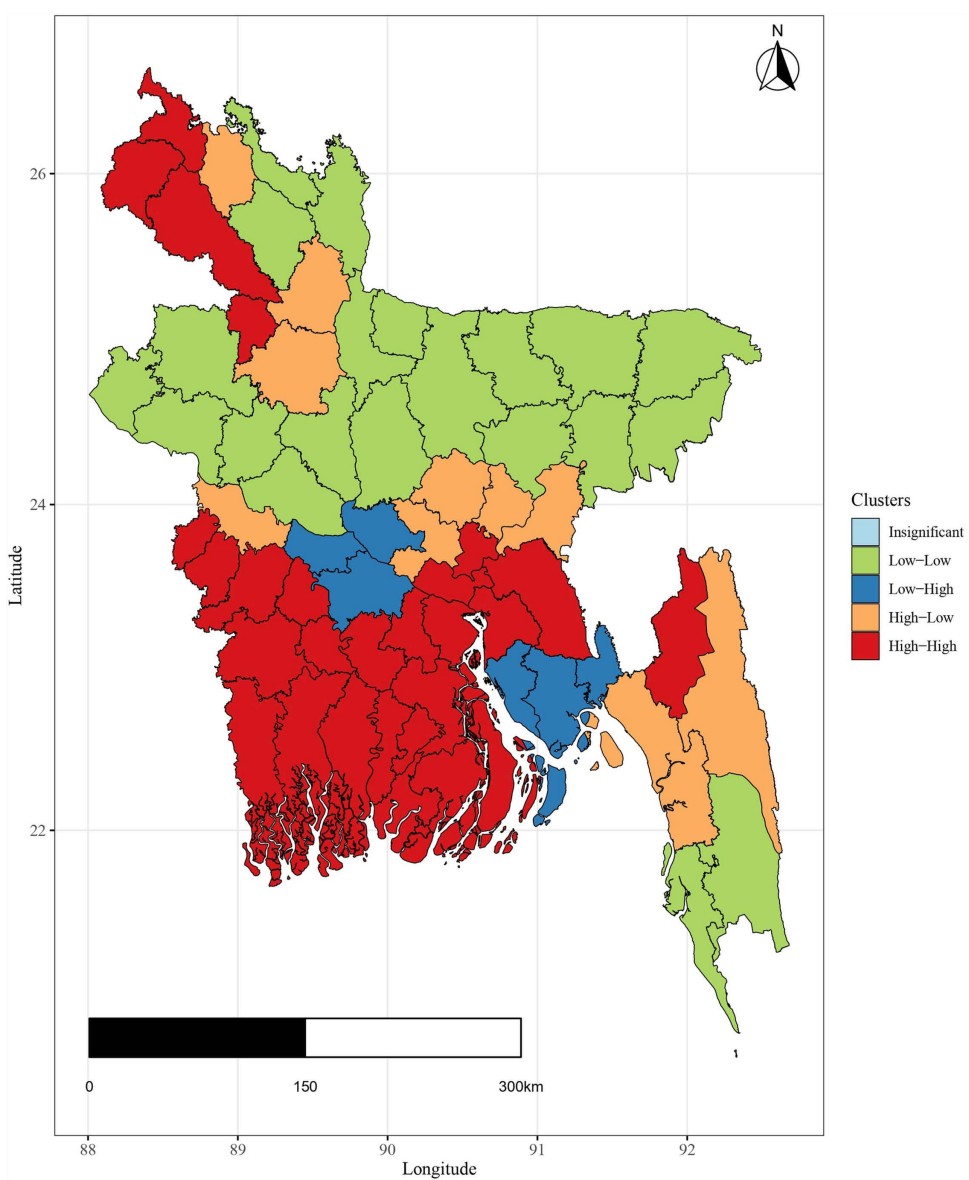

**Fig 12. Spatial clustering (local Moran's *I*) of literacy rate.**

Fig 17 compares the observed and fitted literacy rates over 21 years along with 95% credible intervals. Over the entire period (21 years), the fitted values appear to be fairly balanced. The plot shows neither a major exaggeration nor an underestimate. The lower and upper lines are also pretty near the fitted values, indicating a good accuracy of the AR (1) model for the spatial-temporal framework [Fig 17].

## Discussion

Development disparities among regions are a prevalent occurrence, observed in both developed and developing nations, with more pronounced variations in the latter, such as Bangladesh, where progress unevenly advances across different geographical areas. This article attempts to find out the spatial and temporal variations in educational development (literacy)

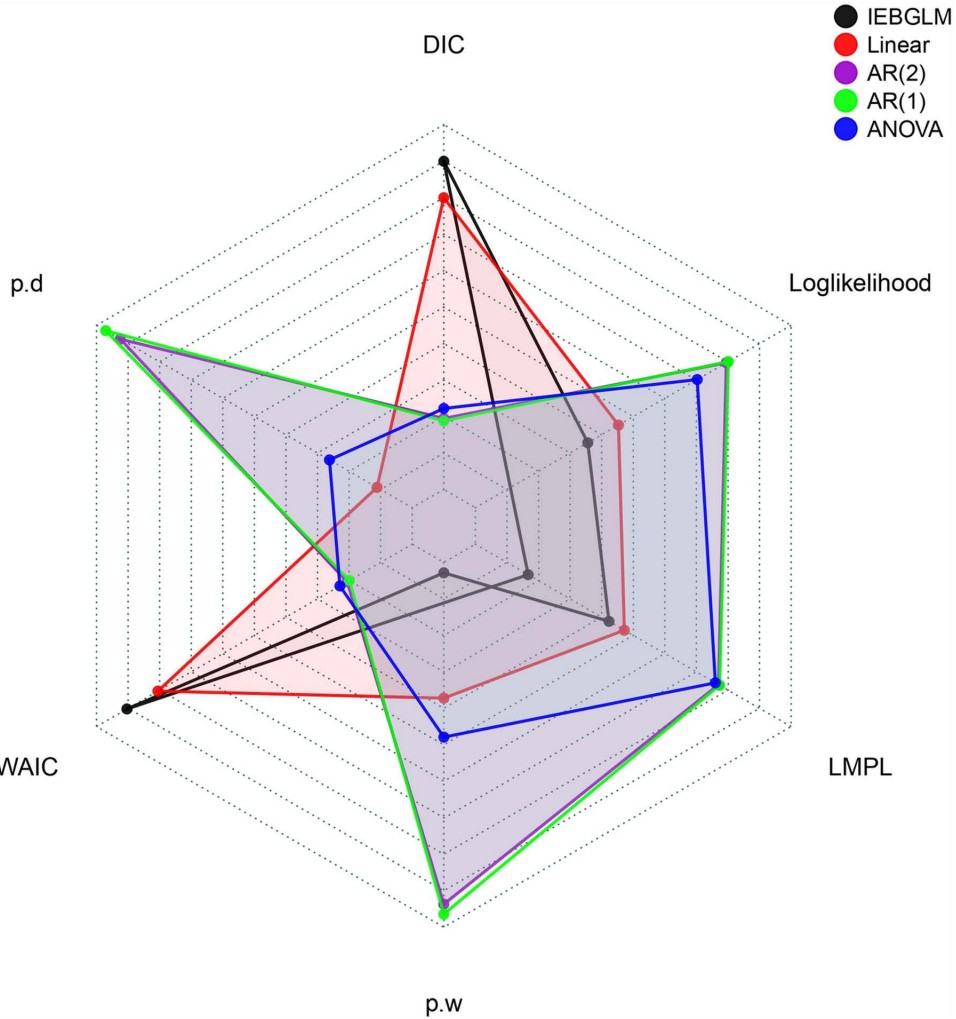

**Fig 13. Comparison plot for DIC measures of the selected models.**

at regional levels in the country. The findings of this study depict that in Bangladesh at the district level, the literacy rate has been growing over the years, but there exists inequity in its rates among districts. A previous study also observed that the literacy rate has increased significantly in Bangladesh to achieve sustainable development goals (SDG)-4 [6]. A vital prerequisite for accelerating the achievement of other Sustainable Development Goals is guaranteeing all students have access to high-quality education [32] and a well-diversified education system [33]. Numerous studies have highlighted the strong connection between socioeconomic status and literacy rates in Bangladesh. Lower-income families often struggle to afford education-related expenses, leading to reduced access to quality education [15,17]. Like, a previous study examined the relationship between poverty and educational attainment [34] this study has highlighted spatial variations in education facilities by identifying developed and lagging regions in the country.

Findings depict that Noakhali, Manikganj, Faridpur, Feni, Lakshmipur, and Rajbari have a low literacy rate, while it would also seem that the districts next to them have a high literacy rate. It is observed that 10 districts give the impression of having a high literacy rate. Moreover, 27 districts are in the high-high cluster category, indicating that they

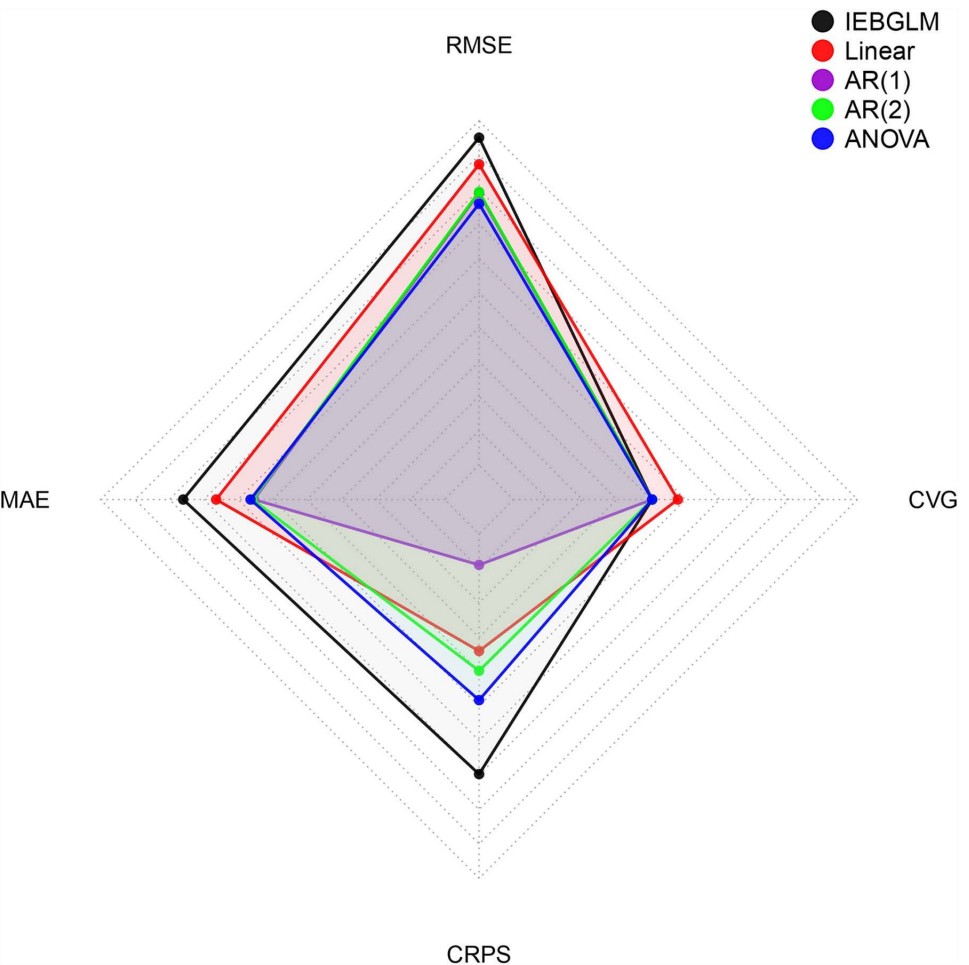

**Fig 14. Comparison plot for RMSE measures of the selected models.**

have neighboring districts containing a high educational development. In this study, it is observed a noteworthy influence of the income index on literacy rates (EDI), affirming that higher family income is associated with elevated levels of literacy. This underscores the reciprocal relationship between income and literacy, indicating that while illiteracy may not be the cause of poverty, poverty indeed contributes to lower literacy levels. The health and nutritional status of individuals, particularly children, can affect cognitive development and, subsequently, their ability to learn and become literate [16]. This study confirms a discernible impact of the health index on the overall literacy rate, highlighting their interconnected relationship. Historically, global literacy rates have significantly increased over the past two centuries, primarily due to higher enrollment in primary education and substantial growth in secondary and tertiary education [18]. Today, average years of schooling are substantially higher compared to a century ago and our study represents the high influence of expected years of schooling on literacy rate. Again, Urban areas tend to have better educational facilities and resources, leading to higher literacy rates compared to rural regions and research often focuses on this [20]. Academic performance is below the national average in several areas of Bangladesh that are marked by remoteness and ecological problems [19]. Geographic disparities in terms of access to schools, transportation, and infrastructure often

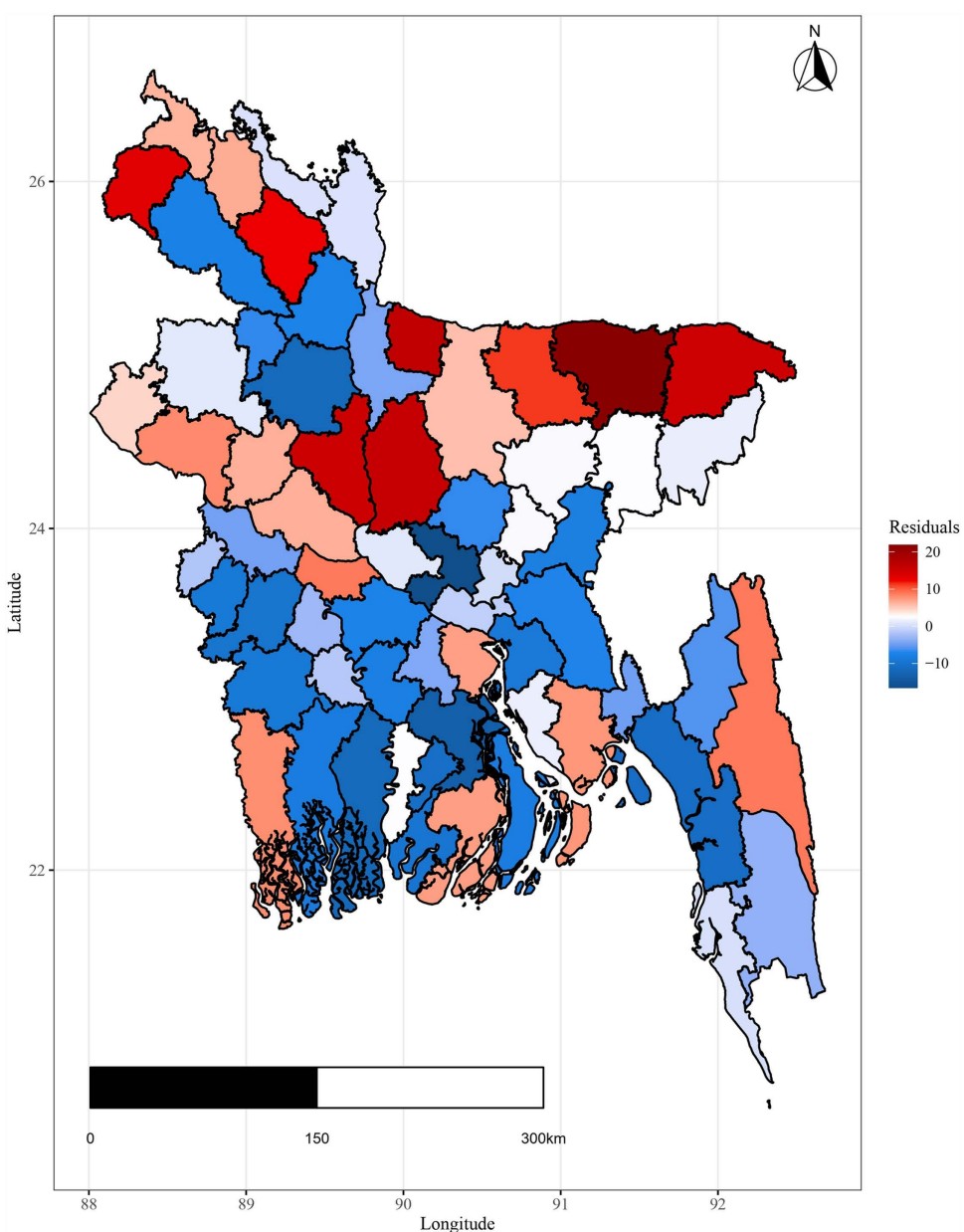

**Fig 15. District-level, spatially aggregated residuals.**

result in lower literacy rates in remote or underdeveloped areas [35]. Inequality in service delivery across adjacent town neighborhoods is discovered as a result, and the overall spatial distribution of the schools exhibits a clustered pattern [11]. A previous study [36] revealed a degree of spatial coherence in the distribution of literacy rates and the presence of educational institutions in Bangladesh. The fitted values indicate the accuracy of the AR (1) model for the spatial-temporal framework and it may be concluded that the model is good enough. The lower and upper lines are also pretty near to the fitted values. This study's findings reaffirm the presence of spatial consistency in the distribution of EDI and disparities in the availability of educational facilities in the country.

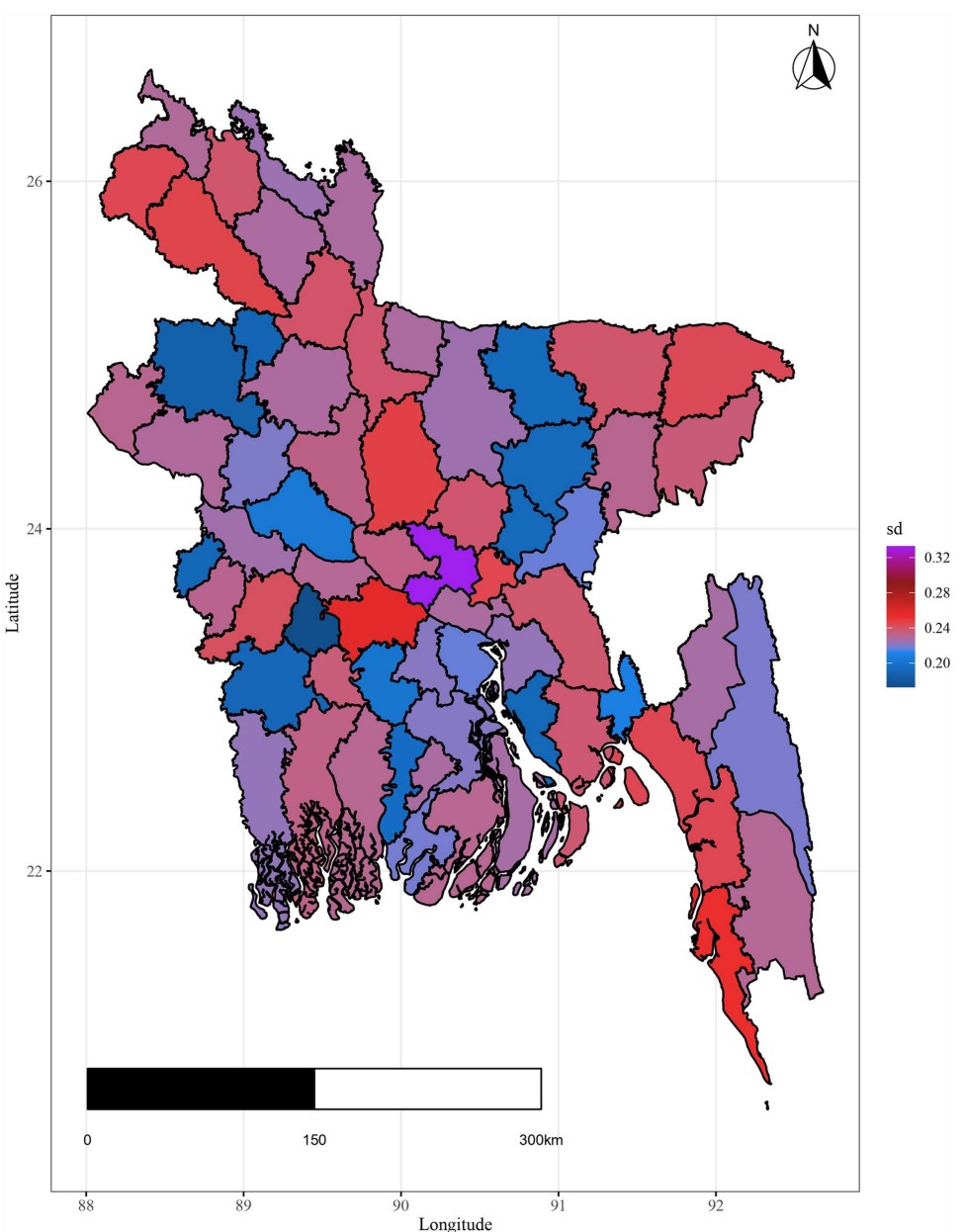

**Fig 16. District-level residual standard error.**

## Conclusion

This study revolves around a comprehensive assessment of the interplay between spatial and temporal dimensions impacting educational development within Bangladesh. Additionally, the authors identified the factors influencing educational development. The findings of our analysis indicate the presence of both spatial and temporal effects on educational development in Bangladesh. Further, we identified that expected years of schooling, health, and income indices exhibit considerable influence on EDI. However, the variable "distance" exerts a relatively minor effect on EDI and the dependency ratio hurts it. The results of this paper suggest that although the government has unique strategies for the districts outside of Dhaka, they are

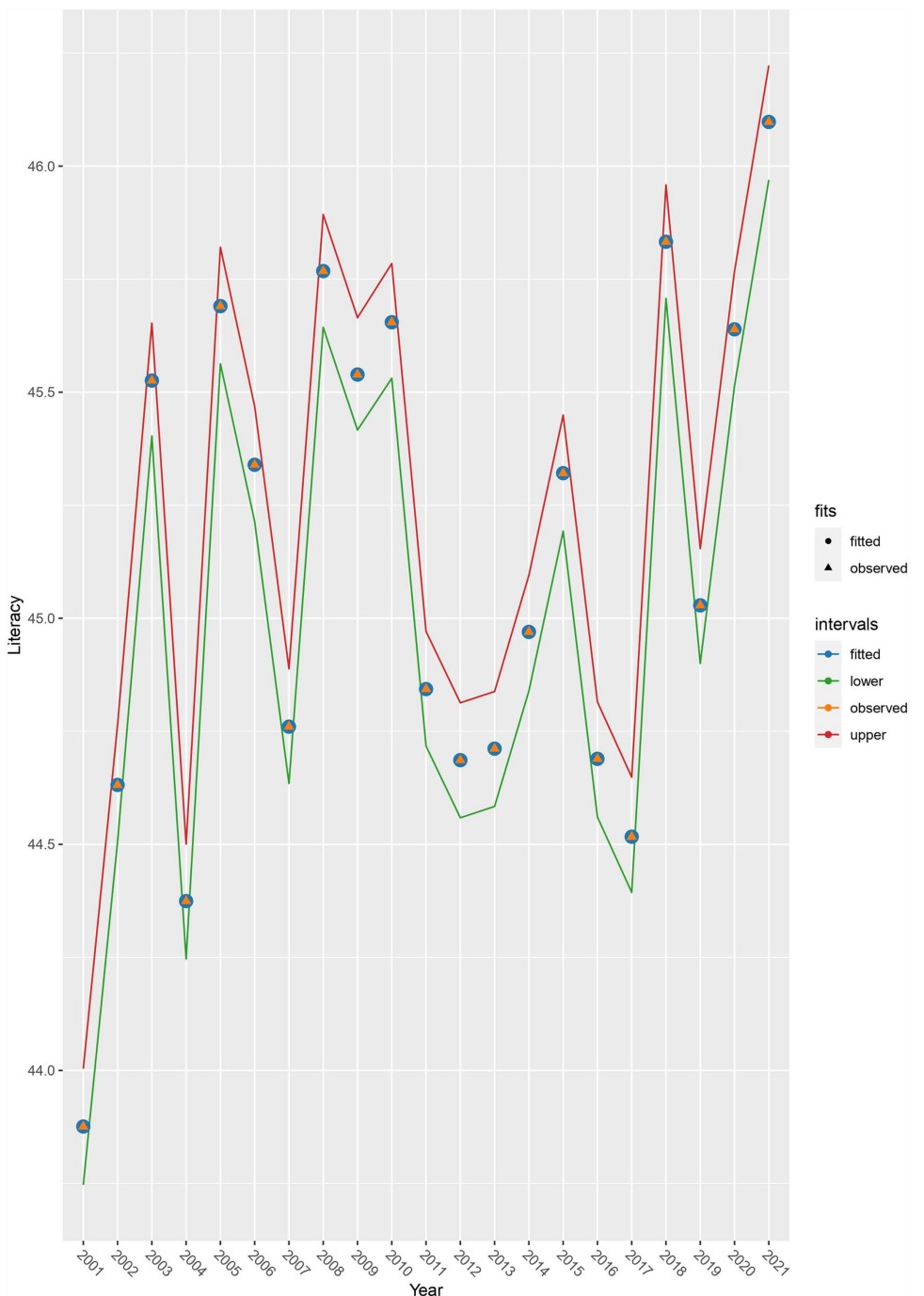

**Fig 17. Observed and fitted time-series plot for literacy rate with 95% credible intervals.**

insufficient. This would enable them to develop the required policies and put development initiatives into practice. This study, by considering both spatial and temporal dimensions, provides policymakers with a holistic grasp of the complex aspects of regional development disparities. In conclusion, the findings point to the necessity of concentrating differently on community-level programs aimed at boosting access to educational opportunities, particularly

in Bangladesh's underprivileged districts. Policymakers should prioritize given initiation which can be recommended from the outcome of our research. Furthermore, the authors are hopeful that the findings will be motivating for the researchers who want to examine how development indices are affected by time and region, which they may use to get a concise assessment of the features of various districts in Bangladesh in terms of development indicators.

## Supporting information

**Data File. Data_EDI.csv.**
(CSV)

## Author contributions

**Conceptualization:** Afroza Sultana, Md. Sifat Ar Salan, Mohammad Alamgir Kabir, Md. Moyazzem Hossain.

**Data curation:** Afroza Sultana, Akher Ali, Md. Sifat Ar Salan.

**Formal analysis:** Afroza Sultana, Md. Sifat Ar Salan.

**Investigation:** Md. Sifat Ar Salan.

**Methodology:** Afroza Sultana, Md. Sifat Ar Salan, Mohammad Alamgir Kabir, Md. Moyazzem Hossain.

**Software:** Afroza Sultana, Md. Sifat Ar Salan.

**Supervision:** Md. Sifat Ar Salan, Mohammad Alamgir Kabir, Md. Moyazzem Hossain.

**Visualization:** Afroza Sultana, Akher Ali.

**Writing – original draft:** Afroza Sultana, Akher Ali, Md. Sifat Ar Salan, Md. Moyazzem Hossain.

**Writing – review & editing:** Mohammad Alamgir Kabir, Md. Moyazzem Hossain.

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
