## [Decision Letter · Decision Letter 0]

14 Aug 2024

Dear Dr. Hossain,

Thank you for submitting your manuscript to PLOS ONE. After careful consideration, we feel that it has merit but does not fully meet PLOS ONE’s publication criteria as it currently stands. Therefore, we invite you to submit a revised version of the manuscript that addresses the points raised during the review process.

We cannot make any decision about publication until we have seen the revised manuscript and your response to thereviewers' comments. Your revised manuscript is also likely to be sent to reviewers for further evaluation. 

We look forward to receiving your revised manuscript.

Kind regards,

Swaminathan Subramanian, Ph.D.

Academic Editor

PLOS ONE

Journal Requirements:

**Additional Editor Comments:**

The researchers examined the spatial and temporal variation in educational development index (EDI) in Bangladesh, aiming to understand the driving factors of EDI. They fitted a Bayesian spatial temporal model to district level literacy rate (I presume it is EDI) assuming literacy rate as Gaussian distributed and adjusting for socio-economic, and demographic characteristics. They identified health index, income index, expected years of schooling, population density, and dependency ratio are important factors of EDI in Bangladesh and concluded that policymakers can use these factors to identify the districts to improve the educational index. The manuscript needs major revision in respects of Introduction. Methods, and presentation of Results. Therefore, in its present form, the ms is not suitable for publication.

**Major** :

The model description is incomplete: The authors state that they have used spatiotemporal models to describe the evolution of EDI over space and time. Whether ANOVA model is same as space-time interaction model as proposed by Knorr and Held (Bayesian modelling of inseparable space-time variation in disease risk. Statistics in medicine. 2000;19(17–18):2555–67. pmid:10960871 2000).  If so whether the authors fitted all the four types of interactions (Type I, II, III, IV). If not, which one of the 4 type of interaction models were fitted to data, giving reason for their choice. What is Autoregressive model? Please make it clear what does the components of the AR model ( and ) describe (spatial or temporal, or space-time interactions). A clear description of the spatial / temporal or spatiotemporal components of each model for ‘*’*  should be given. E.g. What does the symbols refer to ‘’ in ANOVA, and  and  in the autoregressive models. Alternatively, apart from Linear Model trend, I would suggest the authors to provide a generic equation to describe the response variable as a linear combination of fixed effects (explanatory variables), and random effects (spatial, temporal and space-time interaction). This equation can be used to fit only a fixed effect model, or fixed and random effect models (only spatial or temporal, with or without interaction term). The methodology and presentation of results lack clarity, needs improvement. The Methods are described in Introduction as well as in the Results Section. As a consequence, lacks continuity while reading the Results Section. Introduction is too lengthy and is a mixture of Introduction and Methods. The Introduction has lot of scope for shortening. 

**Minor**

Please consider revising the title to reflect the objectives of the ms as given beow:  “Modeling Spatial and Temporal Variability in Educational Development Index of Bangladesh using socio-economic, and demographic data, 2001–2021: A Bayesian Approach”

**ABSTRACT**

The ‘AR1’ model is a ‘temporal model’. Instead, the authors should rewrite the sentence as “Among the several models tested, the temporal model ‘AR1’ model showed better performance compared with others”. Please consider rewriting the sentence starting from “However, the health index,……” as “Of the factors considered for model fitting, the health index, income index, expected years of schooling, population density, and dependency ratio are found to be important components of educational development in Bangladesh.”

**INTRODUCTION**

Please clarify whether EDI and literacy rate mean the same. If so, use the same terminology throughout the ms.Introduction, para 1: delete the expansion of ‘EDI’ as it is already abbreviated and use EDI in the subsequent occurrences.Move the sentences from INTRIDUCTION describing Methodology to “METHODS’ section. E.g. The sentences beginning with “spatial data mining ……” until the end of the para can be moved to Methods section.Last para: Delete the sentences beginning with “The hierarchical…. … educational development data [14].” 

**METHODS**

Data Source: Abbreviate “Global Data Lab” in the first occurrence as “GDL”. Please list the variables for which the data were extracted from this source or simply say all the covariates data were extracted from this source.The authors can include a table showing the study variables, time period and the source for each. Alternatively, they can link the variables with data source. At present the data source is given separately without a reference to the study variables.The probability mass function: The response variable is a continuous one. Therefore, it should be read as “probability density function”. I presume, the equation is erroneously typed, it should have begun with, for e.g. “f(y, mu, sigma)”. Please check and modify the equation accordingly.Please expand “ANOVA”. Ethical statement:  The authors have to justify the reasons for “not applicable”.

**RESULTS **

The Results begins with a description of the ‘EDA’. This should be moved to Methods. Para 3: “To explore the relationship between literacy rate and the other social index, the authors used a correlation matrix presented in Figure 2” Please read the sentence as “Figure 2 shows correlation matrix describing the relationships between literacy rate and socio-economic variables”. Move the sentence, “To examine autocorrelation with Global Moran’s, we set the null hypothesis as, = There is no spatial dependence between neighboring districts due to educational development, and alternative hypothesis  = There is spatial dependence between neighboring districts due to educational development” to Methods section under “Assumption checking”The para beginning with “Exploratory spatial data analysis (ESDA) has difficulties in capturing the temporal dynamics of geographical features, even though it can be used to depict and summarize complicated spatial patterns [32]. To capture temporal dynamics along with spatial effect, Bayesian spatio-temporal.” The para describes the methods. Please move these sentences to appropriate place under Methods Section.Table 3: Add a footnote to Table 3 to expand the abbreviated columns labels (DIC, WAIC, .p.d, p.w, LMPL, RMSE, MAE, CRPS, CVG).‘Figure 7’ should read as ‘Figure 6’

In addition to the above, comments / suggestions are added in the pdf file of the ms as comment tags (Please refer to the attached ms). 

Reviewers' comments:

Reviewer's Responses to Questions

**Comments to the Author**

1. Is the manuscript technically sound, and do the data support the conclusions?

Reviewer #1: Yes

Reviewer #2: Yes

2. Has the statistical analysis been performed appropriately and rigorously?

Reviewer #1: I Don't Know

Reviewer #2: Yes

3. Have the authors made all data underlying the findings in their manuscript fully available?

Reviewer #1: Yes

Reviewer #2: Yes

4. Is the manuscript presented in an intelligible fashion and written in standard English?

Reviewer #1: Yes

Reviewer #2: Yes

Reviewer #1: Dear authors,

This is an interesting paper which contains informative findings and plots.

There is a good balance of exploratory and inferential statistics.

Some areas for improvement are:

1. I would suggest more work on describing the modeling approach and making it "more accesible" to the readership of the journal.

2. Higher resolution of the images would improve the paper.

3. Who is Sahu? This name was used in the text.

4. The section describing the model notation is not accessible to the readership. Would be useful to explain which components of the model account for spatial variation and spatiotemporal ie. improve the signposting for the readership of the journal

5. Before using abbreviations, such as bmstdr it is best to describe what they mean at the first mention.

6. The Discussion should sumarrize what the spatiotemporal effects as well as the spatial and temporal effects. Point estimates and Credible intervals would be useful.

Hope these are useful to you.

Reviewer #2: In this paper, Sultana et al. aim to examine how the educational development index relates to various spatiotemporal variables. This study is based on secondary data on literacy rates from 64 districts of Bangladesh and 6 relevant variables over the period 2001 to 2021. The optimal model for the data was identified from Bayesian spatial-temporal modeling (Linear, ANOVA, AR1, and AR2) and the Markov Chain Monte Carlo (MCMC) method used to generate data about the prior and posterior realizations.

The work is well done. I have some specific questions.

1. Page 5. (_, \mu_, \sigma^_) should be (_ | \mu_, \sigma^_). In this formula, what about \sigma^2_it? Are these fixed?

2. In all the maps, there is a figure legend “0,0.5, 1m”. why include this? What is “m”? suggest removal.

3. Page 6. “Conditional Autoregressive (CAR)” needs a citation.

4. Figure 5. What clustering method is used?

5. Figure 7. It is very difficult to distinguish circles representing “fitted” and triangle representing “observed”.

6. Table 2. Intercept. CI should be kept in one line. Reduce font size.

7. Table 3. Please indicate which model is the best.

**Do you want your identity to be public for this peer review?** For information about this choice, including consent withdrawal, please see our Privacy Policy

Reviewer #1: No

Reviewer #2: No

---

## [Author Response · Author response to Decision Letter 1]

25 Sep 2024

Author responses to the review comments:

We would like to express our sincere gratitude to the two reviewers and the Editor as well as in-house editor for their valuable comments. We have considered all the comments made by the reviewers and thoroughly revised and formatted the manuscript accordingly. A detailed response to each of the comments is provided ibelow:

Author's Response to Editor comments:

Thank you very much for completing the review process and providing the comments and feedback. We believe that it helps to improve the quality of the manuscript.

We realize the pain of managing the reviewers. As an academic editor, I have a similar experience. We revised the manuscript carefully based on the review comments.

Yes, you may take the decision following journal requirements as well as your own way. Revised texts are in red colour.

Thank you very much. We revised the manuscript accordingly and uploaded the required files to the journal system.

Author's Response to Journal Requirements:

1. Thank you. We revised the manuscript following the PLOS ONE style. Revised texts are in red colour.

Page: 1-21

2. Thanks. We add the code-sharing statement to the revised manuscript. Revised texts are in red colour.

Page: 20

Author's Response to Additional Editor comments:

Thank you very much for your insightful comments and feedback. We believe that it helps to improve the quality of the manuscript. We revised the manuscript as per review comments. Revised texts are in red colour.

Author's Response to Additional Editor comments:

1. Thank you very much for your valuable comment. We have updated the manuscript by improving the methodology section as well as adding model description. We clarify that the ANOVA model we employed is distinct from the space-time interaction model by Knorr and Held (2000) and does not specifically address their four types of interactions. Our autoregressive model (AR) includes components for both spatial and temporal dependencies, but we did not implement a full space-time interaction term as AR (1) is temporal autoregressive with δ_t=0. We have revised the manuscript to include a generic equation describing the response variable as a linear combination of fixed effects (explanatory variables) and random effects (spatial, temporal, and space-time interactions) to better explain our modelling approach. Revised texts are in red colour.

Page: 2, 5, 7

2. Thanks a lot for providing this most important comment. The methodology and presentation of results have been improved by doing some corrections. Some of the content of the introductions and results have been replaced in the methodology section of this study as well as updated the result section to make clarity of the study. Revised texts are in red colour.

Page: 2-3, 5-6

3. Thank you for your observation regarding the length and content of the Introduction. We agree that it contained elements more appropriate for the Methods section and was overly detailed. We have now revised the Introduction to focus solely on providing the necessary background, objectives, and significance of the study while moving methodological details to the appropriate methods section. This has resulted in a more concise and focused Introduction that better aligns with the manuscript's overall structure. Revised texts are in red colour.

Page: 2-3

4. We are grateful for your comments. We have changed the title with respect to your suggestion to make it more significant. Revised texts are in red colour.

Page: 1

5. Thank you for your observation regarding the ‘AR1’ model is a ‘temporal model’ and we have rewritten the sentence according yours. Revised texts are in red colour.

Page: 2

6. We appreciate your insightful comment. We considered rewriting the sentence starting from “However, the health index,……” as “Of the factors considered for model fitting, the health index, income index, expected years of schooling, population density, and dependency ratio are found to be important components of educational development in Bangladesh.” Revised texts are in red colour.

Page: 2

7. Thanks. Yes, EDI and literacy rates are the same and according to your consent, we have used the same terminology throughout the manuscript.

8. Thank you so much for your comments. We have deleted the expansion of “EDI” from the introduction section. Revised texts are in red colour.

Page: 2

9. Thanks a lot for your queries. We have moved the beginning sentence to the Methods section. Revised texts are in red colour.

Page: 4

10. Thanks a lot for your suggestion. We have deleted the sentence from the last paragraph. Revised texts are in red colour.

Page: 4

11. Thanks for your insightful comments. The variables included in this study is discussed in the Study Variables section.

Revised texts are in red colour.

Page: 4-5

12. Thank you so much. According to your alternative comment, we have added citations of the links to the data sources. Revised texts are in red colour.

Page: 4

13. Thanks a lot for your comments. We have changed the probability mass function to the probability density function. We have checked and corrected the equation to the true form. Revised texts are in red colour.

Page: 6

14. Thanks. We expended the ANOVA. Revised texts are in red colour.

Page: 7

15. Thank you so much. We have justified the ethical statement section which is located before the results section. Revised texts are in red colour.

Page: 8

16. Thank you. The description of the “EDA” has been moved to the method section. Revised texts are in red colour.

Page: 5, 8

17. Thank you so much. We have updated the manuscript to add the sentence as “Fig 2 shows correlation matrix describing the relationships between literacy rate and socio-economic variables”. Revised texts are in red colour.

Page: 8

18. Thank you for pointing this out. We have moved the sentence, “To examine autocorrelation with Global Moran’s, we set the null hypothesis as, = There is no spatial dependence between neighbouring districts due to educational development, and alternative hypothesis = There is spatial dependence between neighbouring districts due to educational development,” to the Methods section under the “Assumption Checking” subsection. Revised texts are in red colour.

Page: 6

19. Thank you for your feedback. We have moved the paragraph beginning with “Exploratory spatial data analysis (ESDA) has difficulties in capturing the temporal dynamics of geographical features, even though it can be used to depict and summarize complicated spatial patterns...” to the Methods section, where it is more appropriate. Revised texts are in red colour.

Page: 4

20. Thank you for your suggestion. We have added a footnote to Table 3 to expand the abbreviated column labels. Revised texts are in red colour.

Page: 15

21. Thank you for pointing out the labeling error. We have corrected the labeling of all figures. Revised texts are in red colour.

Page: 8,9,10,11,12,16, 17

Author's Response to Reviewer 1 Comments:

1. Thank you for your valuable suggestion. We have revised the manuscript to provide a clearer and more detailed description of the modeling approach. We have simplified technical language where possible and included additional explanations to make the concepts more accessible to a broader readership. This includes breaking down complex terms, providing intuitive explanations of the models used, and ensuring that the rationale behind our choices is clearly communicated. These changes aim to enhance the understanding and accessibility of the methodology for all readers of the journal. Revised texts are in red colour.

Page: 4,5, 6

2. Thank you so much for your insightful comments. We produced high resolution pictures according to your comment and all the pictures are attached in a separate file.

3. Thank you. Sujit Sahu is the author of the book entitled “Bayesian modelling of spatio-temporal data with R” published by Chapman and Hall/CRC Press.

We add the citation using a number instead of author’s name.

4. Thank you for your insightful feedback. We have revised the section describing the model notation to make it more accessible to the readership. Specifically, we have added clear explanations of which components of the model account for spatial variation (e.g.) and spatiotemporal interactions (e.g.) We have also improved signposting throughout the section, providing more intuitive descriptions to help readers better understand how each component contributes to the overall model. These revisions aim to ensure that the model’s structure and its implications for spatial and spatiotemporal analysis are clearly communicated. Revised texts are in red colour.

Page: 5-6

5. Thank you so much for your comment. Here, “bmstdr” is a R package for fitting the spatio-temporal modelling. Revised texts are in red colour.

Page: 7

6. Thank you for your suggestion. We have revised the Discussion section to include a summary of the spatiotemporal effects as well as the individual spatial and temporal effects, which were previously described in the methodology part. Revised texts are in red colour.

Page: 18-19

Author's Response to Reviewer 2 Comments:

1. Thank you so much. We have made these changes by incorporating conditional. You got it exactly. Revised texts are in red colour.

Page: 6

2. Thanks a lot. We revised the scale on the maps.

3. Thank you for your suggestion. We have included the reference of the Conditional Autoregressive (CAR) model. Revised texts are in red colour.

Page: 7

4. Thank you for your query. This is not similar to usual clustering. It is basically based on Moran's I statistic. The clustering method used in Figure 5 is based on Local Moran's I. We have clarified this in the figure caption to ensure that the methodology is clearly communicated to the readers. Clustering in spatial modeling is used to identify patterns and group similar data points based on geographic location, simplifying analysis and revealing spatial relationships. It aids in understanding complex spatial data and improving decision-making processes. Revised texts are in red colour.

Page: 12

5. Thank you so much for your insightful comment. We have changed the picture to distinguish the circle and triangle. Actually, there is a very small deviation between fitted and observed values. Revised texts are in red colour.

Page: 17

6. Thank you for your comment. We have updated the Table 2. Revised texts are in red colour.

Page: 13

7. Thank you for your suggestion. We have updated Table 3 to clearly indicate which model is the best based on the performance criteria (e.g., lowest DIC, WAIC, RMSE, etc.). The best-performing model is now highlighted in the table and texts. Revised texts are in red colour.

Page: 15

Author's Response to Reviewer Comments in PDF:

Thanks. We revised it. Revised texts are in red colour.

Page: 1

Thanks. We revised the manuscript as per the review comments. Revised texts are in red colour.

Page: 4-5

Thank you. We deleted the sentence. Revised texts are in red colour.

Page: 4

Thanks. We revised it. Revised texts are in red colour.

Page: 4-5

Thanks. We revised the manuscript accordingly. Revised texts are in red colour.

Page: 8

We appreciate your comment. We revised the manuscript accordingly. Revised texts are in red colour.

Page: 6

Thank you. We revised the manuscript as per your comments. Revised texts are in red colour.

Page: 4-5

Thank you. We revised the manuscript as per your comments. Revised texts are in red colour.

Page: 4-5

Thanks. We add the abbreviations. Revised texts are in red colour.

Page: 15

Thanks. We corrected the typo. Revised texts are in red colour.

Page: 15-16

Finally, the revised manuscript has been produced following the valuable comments and suggestions of the reviewers. Once again, we would like to thank the reviewers for their sincere dedication, professional insights, and earnest cooperation in reviewing the manuscript.

---

## [Decision Letter · Decision Letter 1]

5 Dec 2024

Dear Dr. Hossain,

Thank you for submitting your manuscript to PLOS ONE. After careful consideration, we feel that it has merit but does not fully meet PLOS ONE’s publication criteria as it currently stands. Therefore, we invite you to submit a revised version of the manuscript that addresses the points raised during the review process.

I have made substantial editorial corrections in the attached ms, particularly the model descriptions under Methodology, and a few corrections in the Results, Discussion and Conclusions. I request you go through the editorial corrections in the above-mentioned sections and revise the text, if they are agreeable. Also, you will find a few comments in respect of model description and Results for your consideration while revising the ms.

We look forward to receiving your revised manuscript.

Kind regards,

Swaminathan Subramanian, Ph.D.

Academic Editor

PLOS ONE

Journal Requirements:

Additional Editor Comments:

The following are a few comments to be addressed while revising ms:

1) LL 66: Use the PLOS ONE style for citing references.

2) LL 44-49 & 77-78: This is a repeat statement, should be deleted. Please see

3) LL 169, AR model equation: Is it ‘ϕ_it’ or ‘ϕ_i’. Else define ‘ϕ_it’ in the AR model

4) LL 169 & 453-54: ANOVA model: Please describe the parameter (γ_it) in the ANOVA model. What does this parameter represent?

5) LL 580 and Table 2: Is it ν^2 or σ^2. ν^2 is no where described in the model equations. Please check and correct.

6) LL 581-85: The estimated values for these parameters are not matching the data provided in Table 2. Please check and revise the text / Table.

7) Table 2: 'τ^2_int and τ^2_slo': No data given for these parameters. Please check the table and revise it.

8) LL 611-12: You have not considered the joint effect of space and time. Explain, how did you make this inference.

Reviewers' comments:

Reviewer's Responses to Questions

**Comments to the Author**

Reviewer #1: All comments have been addressed

Reviewer #2: All comments have been addressed

2. Is the manuscript technically sound, and do the data support the conclusions?

Reviewer #1: Yes

Reviewer #2: Yes

3. Has the statistical analysis been performed appropriately and rigorously?

Reviewer #1: I Don't Know

Reviewer #2: Yes

4. Have the authors made all data underlying the findings in their manuscript fully available?

Reviewer #1: Yes

Reviewer #2: Yes

5. Is the manuscript presented in an intelligible fashion and written in standard English?

Reviewer #1: Yes

Reviewer #2: Yes

Reviewer #1: The issues that I raised are adequately addressed. For the more technical queries by other reviewers, I am not sure of whether they have been addressed.

Reviewer #2: The authors have addressed all my comments.

**Do you want your identity to be public for this peer review?** For information about this choice, including consent withdrawal, please see our Privacy Policy

Reviewer #1: No

Reviewer #2: No

---

## [Author Response · Author response to Decision Letter 2]

16 Dec 2024

Author responses to the review comments:

We would like to express our sincere gratitude to the two reviewers and the Editor for their valuable comments. We have considered all the comments made by the reviewers and thoroughly revised and formatted the manuscript accordingly. A detailed response to each of the comments is provided below.

Author's Response Editor Comments:

Thank you very much for your careful checking and revision. We appreciate it. We also revised the manuscript as per your comments. Revised texts are in red colour.

Thank you very much. We revised the manuscript accordingly and uploaded the required files to the journal system.

Author's Response Journal Requirements:

Thank you very much. We checked all references and confirm that all are correct.

Author's Response Additional Editor comments:

Thanks. We revised it. Revised texts are in red colour.

Page: 3

Thanks. We deleted the repeated statement. Revised texts are in red colour.

Page: 3

Thank you very much. We revised it. Revised texts are in red colour.

Page: 5

Thanks. We revised the manuscript. Revised texts are in red colour.

Page: 6

Thanks. We revised the manuscript. Revised texts are in red colour.

Page: 13-14

Thanks. We revised the Table 2. Revised texts are in red colour.

Page: 12-13

Thanks. There were typos. We revised the Table 2. Revised texts are in red colour.

Page: 13

Thanks for your careful checking. We deleted this sentence. Revised texts are in red colour.

Page: 15

Finally, the revised manuscript has been produced following the valuable comments and suggestions of the reviewers. Once again, we would like to thank the reviewers for their sincere dedication, professional insights, and earnest cooperation in reviewing the manuscript.

---

## [Editor Report · Decision Letter 2]

20 Dec 2024

Modeling Spatial and Temporal Variability in Educational Development Index of Bangladesh using socio-economic, and demographic data, 2001–2021: A Bayesian Approach

PONE-D-24-14527R2

Dear Dr. Hossain,

We’re pleased to inform you that your manuscript has been judged scientifically suitable for publication and will be formally accepted for publication once it meets all outstanding technical requirements.

Kind regards,

Swaminathan Subramanian, Ph.D.

Academic Editor

PLOS ONE
---

## [Editor Report · Acceptance letter]

PONE-D-24-14527R2

PLOS ONE

Dear Dr. Hossain,

I'm pleased to inform you that your manuscript has been deemed suitable for publication in PLOS ONE. Congratulations! Your manuscript is now being handed over to our production team.

Kind regards,

on behalf of

Dr. Swaminathan Subramanian

Academic Editor

PLOS ONE